# Practical and Asymptotically Exact Conditional Sampling in Diffusion Models

**Luhuan Wu**[*]
Columbia University
lw2827@columbia.edu

**Brian L. Trippe**[*]
Columbia University
blt2114@columbia.edu

**Christian A. Naesseth**
University of Amsterdam
c.a.naesseth@uva.nl

**David M. Blei**
Columbia University
david.blei@columbia.edu

**John P. Cunningham**
Columbia University
jpc2181@columbia.edu

## Abstract

Diffusion models have been successful on a range of conditional generation tasks including molecular design and text-to-image generation. However, these achievements have primarily depended on task-specific conditional training or error-prone heuristic approximations. Ideally, a conditional generation method should provide exact samples for a broad range of conditional distributions without requiring task-specific training. To this end, we introduce the *Twisted Diffusion Sampler*, or *TDS*. TDS is a sequential Monte Carlo (SMC) algorithm that targets the conditional distributions of diffusion models through simulating a set of weighted particles. The main idea is to use *twisting*, an SMC technique that enjoys good computational efficiency, to incorporate heuristic approximations without compromising asymptotic exactness. We first find in simulation and in conditional image generation tasks that TDS provides a computational statistical trade-off, yielding more accurate approximations with many particles but with empirical improvements over heuristics with as few as two particles. We then turn to motif-scaffolding, a core task in protein design, using a TDS extension to Riemannian diffusion models; on benchmark tasks, TDS allows flexible conditioning criteria and often outperforms the state-of-the-art, conditionally trained model.[2]

## 1 Introduction

Conditional sampling is an essential primitive in the machine learning toolkit. One begins with a generative model that parameterizes a distribution $p_\theta(x)$ on data $x$, and then augments the model to include information $y$ in a joint distribution $p_\theta(x, y) = p_\theta(x)p(y \mid x)$. This joint distribution then implies a conditional distribution $p_\theta(x \mid y)$, from which desired outputs are sampled.

For example, in protein design, $p_\theta(x)$ can represent a distribution of physically realizable protein structures, $y$ a substructure that imparts a desired biochemical function, and samples from $p_\theta(x \mid y)$ are then physically realizable structures that contain the substructure of interest [e.g. 34, 37].

Diffusion models are a class of generative models that have demonstrated success in conditional generation tasks [18, 27, 33, 37]. They parameterize distributions $p_\theta(x)$ through an iterative refinement process that builds up data from noise. When a diffusion model is used for conditional generation, this refinement process is modified to account for conditioning at each step [9, 19, 27, 33].

---

[*]Equal contribution, order by coin flip.

[2]Code: `https://github.com/blt2114/twisted_diffusion_sampler`

37th Conference on Neural Information Processing Systems (NeurIPS 2023).

One approach is to incorporate the conditioning information into training [e.g. 9, 27]; one either modifies the unconditional model to take $y$ as input or trains a separate conditional model to predict $y$ from partially noised inputs. However, conditional training requires (i) assembling a large set of paired examples of the data and conditioning information $(x, y)$, and (ii) designing and training a task-specific model when adapting to new conditioning tasks. For example, image inpainting and class-conditional image generation can be both formalized as conditional sampling problems based on the same (unconditional) image distribution $p_\theta(x)$; however, the conditional training approach requires training separate models on two curated sets of conditioning inputs.

To avoid conditional training, a separate line of work uses heuristic approximations that directly operate on unconditional diffusion models: once an unconditional model $p_\theta(x)$ is trained, it can be flexibly combined with various conditioning criteria to generate customized outputs. These approaches have been applied to *inpainting* problems [24, 33, 34], and other inverse problems [1, 6, 19, 32]. But it is unclear how well these heuristics approximate the exact conditional distributions they are designed to mimic; for example on inpainting tasks they often fail to return outputs consistent with both the conditioning information and unconditional model [40]. These concerns are critical in domains that require accurate conditionals. In molecular design, for example, even a small approximation error could result in atomic structures with chemically implausible bond distances.

This paper develops a practical and asymptotically exact method for conditional sampling from an unconditional diffusion model. We use sequential Monte Carlo (SMC), a general tool for asymptotically exact inference in sequential probabilistic models [5, 10, 25]. SMC simulates an ensemble of weighted trajectories, or *particles*, through a sequence of proposals and weighting mechanisms. These weighted particles then form an asymptotically exact approximation to a desired target distribution.

The premise of this work is to recognize that the sequential structure of diffusion models permits the application of SMC for sampling from conditional distributions $p_\theta(x \mid y)$. We design an SMC algorithm that leverages *twisting*, an SMC technique that modifies proposals and weighting schemes to approach the optimal choices [15, 38]. While optimal twisting is intractable, we effectively approximate it with recent heuristic approaches to conditional sampling [eg. 19], and correct the errors by the weighting mechanisms. The resulting algorithm maintains asymptotic exactness to $p_\theta(x \mid y)$, and empirically it can outperform heuristics alone even with just two particles.

We summarize our contributions: (i) We propose a practical SMC algorithm, *Twisted Diffusion Sampler* or TDS, for asymptotically exact conditional sampling from diffusion models; (ii) We show that TDS applies to a range of conditional generation problems, and extends to Riemannian manifold diffusion models; (iii) On MNIST inpainting and class-conditional generation tasks we demonstrate TDS's empirical improvements beyond heuristic approaches; and (iv) On protein motif-scaffolding problems with short scaffolds TDS provides greater flexibility and achieves higher success rates than the state-of-the-art conditionally trained model.

## 2 Background: Diffusion models and sequential Monte Carlo

**Diffusion models.** A diffusion model generates a data point $x^0$ by iteratively refining a sequence of noisy data $x^t$, starting from pure noise $x^T$. This procedure parameterizes a distribution of $x^0$ as the marginal of a length $T$ Markov chain

$$p_\theta(x^0) = \int p(x^T) \prod_{t=1}^{T} p_\theta(x^{t-1} \mid x^t) dx^{1:T}, \tag{1}$$

where $p(x^T)$ is an easy-to-sample noise distribution, and each $p_\theta(x^{t-1} \mid x^t)$ is the transition distribution defined by the $(T-t)^{\text{th}}$ refinement step.

Diffusion models $p_\theta$ are fitted to match a data distribution $q(x^0)$ from which we have samples. To achieve this goal, a *forward process* $q(x^0) \prod_{t=1}^{T} q(x^t \mid x^{t-1})$ is set to gradually add noise to the data, where $q(x^t \mid x^{t-1}) = \mathcal{N}\left(x^t; x^{t-1}, \sigma^2\right)$, and $\sigma^2$ is a positive variance. To fit a diffusion model, one finds $\theta$ such that $p_\theta(x^{t-1} \mid x^t) \approx q(x^{t-1} \mid x^t)$, which is the reverse conditional of the forward process. If this approximation is accomplished for all $t$, and if $T\sigma^2$ is big enough that $q(x^T)$ may be approximated as $q(x^T) \approx \mathcal{N}(0, T\sigma^2) =: p(x^T)$, then we will have $p_\theta(x^0) \approx q(x^0)$.

In particular, when $\sigma^2$ is small enough then the reverse conditionals of $q$ are approximately Gaussian,

$$q(x^{t-1} \mid x^t) \approx \mathcal{N}\left(x^{t-1}; x^t + \sigma^2 \nabla_{x^t} \log q(x^t), \sigma^2\right), \tag{2}$$

where $q(x^t) = \int q(x^0)q(x^t \mid x^0)dx^0$ and $\nabla_{x^t} \log q(x^t)$ is known as the (Stein) score [33]. To mirror eq. (2), diffusion models parameterize $p_\theta(x^{t-1} \mid x^t)$ via a *score network* $s_\theta(x^t, t)$

$$p_\theta(x^{t-1} \mid x^t) := \mathcal{N}\left(x^{t-1}; x^t + \sigma^2 s_\theta(x^t, t), \sigma^2\right). \tag{3}$$

When $s_\theta(x^t, t)$ is trained to approximate $\nabla_{x^t} \log q(x^t)$, we have $p_\theta(x^{t-1} \mid x^t) \approx q(x^{t-1} \mid x^t)$.

Notably, approximating the score is equivalent to learning a *denoising* neural network $\hat{x}_\theta(x^t, t)$ to approximate $\mathbb{E}_q[x^0 \mid x^t]$. The reason is that by Tweedie's formula [12, 28] $\nabla_{x^t} \log q(x^t) = (\mathbb{E}_q[x^0 \mid x^t] - x^t)/t\sigma^2$ and one can set $s_\theta(x^t, t) := (\hat{x}_\theta(x^t, t) - x^t)/t\sigma^2$. The neural network $\hat{x}_\theta(x^t, t)$ may be learned by denoising score matching (see [e.g. 18, 35]). For the remainder of paper we drop the argument $t$ in $\hat{x}_\theta$ and $s_\theta$ when it is clear from context.

Appendix A generalizes the formulation above to diffusion process formulations that are commonly used in practice.

**Sequential Monte Carlo.** Sequential Monte Carlo (SMC) is a general tool to approximately sample from a sequence of distributions on variables $x^{0:T}$, terminating at a final target of interest [5, 10, 14, 25]. SMC approximates these targets by generating a collection of $K$ *particles* $\{x_k^t\}_{k=1}^K$ across $T$ steps of an iterative procedure. The key ingredients are proposals $r_T(x^T)$, $\{r_t(x^t \mid x^{t+1})\}_{t=0}^{T-1}$ and weighting functions $w_T(x^T)$, $\{w_t(x^t, x^{t+1})\}_{t=0}^{T-1}$. At the initial step $T$, one draws $K$ particles of $x_k^T \sim r_T(x^T)$ and sets $w_k^T := w_T(x_k^T)$, and sequentially repeats the following for $t = T-1, \ldots, 0$:

- *resample*  $\{x_k^{t+1:T}\}_{k=1}^K \sim \text{Multinomial}(\{x_k^{t+1:T}\}_{k=1}^K; \{w_k^{t+1}\}_{k=1}^K)$
- *propose*  $x_k^t \sim r_t(x^t \mid x_k^{t+1}), \quad k = 1, \cdots, K$
- *weight*  $w_k^t := w_t(x_k^t, x_k^{t+1}), \quad k = 1, \cdots, K$

The proposals and weighting functions together define a sequence of *intermediate target* distributions,

$$\nu_t(x^{0:T}) := \frac{1}{\mathcal{L}_t}\left[r_T(x^T) \prod_{t'=0}^{T-1} r_{t'}(x^{t'} \mid x^{t'+1})\right]\left[w_T(x^T) \prod_{t'=t}^{T-1} w_{t'}(x^{t'}, x^{t'+1})\right] \tag{4}$$

where $\mathcal{L}_t$ is a normalization constant. A classic example that SMC applies is the state space model [25, Chapter 1] that describes a distribution over a sequence of latent states $x^{0:T}$ and observations $y^{0:T}$. Each intermediate target $\nu_t$ is constructed to be the posterior $p(x^{0:T} \mid y^{t:T})$ given the first $T - t + 1$ observations, and the final target $p(x^{0:T} \mid y^{0:T})$ is the posterior given all observations.

The defining property of SMC is that the weighted particles at each $t$ form discrete approximations $(\sum_{k'=1}^K w_{k'}^t)^{-1} \sum_{k=1}^K w_k^t \delta_{x_k^t}(x^t)$ (where $\delta$ is a Dirac measure) to $\nu_t(x^t)$ that become arbitrarily accurate in the limit that many particles are used [5, Proposition 11.4]. So, by choosing $r_t$ and $w_t$ so that $\nu_0(x^0)$ matches the desired distribution, one can guarantee arbitrarily low approximation error in the large compute limit.

# 3 Twisted Diffusion Sampler: SMC sampling for diffusion model conditionals

Consider conditioning information $y$ associated with a given likelihood function $p_{y|x^0}(y|x^0)$. We embed $y$ in a joint model over $x^{0:T}$ and $y$ as $p_\theta(x^{0:T}, y) = p_\theta(x^{0:T})p_{y|x^0}(y|x^0)$, where $y$ and $x^{1:T}$ are conditionally independent given $x^0$. Our goal is to sample from the conditional $p_\theta(x^0 \mid y)$.

In this section, we develop Twisted Diffusion Sampler (TDS), a practical SMC algorithm targeting $p_\theta(x^0 \mid y)$. First, we describe how the Markov structure of diffusion models permits a factorization of an extended conditional distribution to which SMC applies. Then, we show how a diffusion model's denoising predictions support the application of *twisting*, an SMC technique in which one uses proposals and weighting functions that approximate the "optimal" ones. Lastly, we extend TDS to certain "inpainting" problems where $p_{y|x^0}(y|x^0)$ is not smooth, and to Riemannian diffusion models.

## 3.1 Conditional diffusion sampling as an SMC procedure

The Markov structure of the diffusion model permits a factorization that is recognizable as the final target of an SMC algorithm. We write the conditional distribution, extended to include $x^{1:T}$, as

$$p_\theta(x^{0:T} \mid y) = \frac{p_\theta(x^{0:T}, y)}{p_\theta(y)} = \frac{1}{p_\theta(y)}\left[p(x^T) \prod_{t=0}^{T-1} p_\theta(x^t \mid x^{t+1})\right]p_{y|x^0}(y|x^0) \tag{5}$$

**Algorithm 1:** Twisted Diffusion Sampler (TDS)

---

1 **for** $k = 1 : K$

2     $x_k^T \sim p(x^T), \quad w_k \leftarrow \tilde{p}_k^T = \tilde{p}_\theta(y \mid x_k^T)$    `// initial proposal and weight`

3 **for** $t = T - 1, \cdots, 0$

4     $\{x_k^{t+1}, \tilde{p}_k^{t+1}\}_{k=1}^K \sim \text{Multinomial}\left(\{x_k^{t+1}, \tilde{p}_k^{t+1}\}_{k=1}^K; \{w_k\}_{k=1}^K\right)$    `// resample`

5     **for** $k = 1 : K$

6        $\tilde{s}_k = (\hat{x}_\theta(x_k^{t+1}) - x_k^{t+1})/t\sigma^2 + \nabla_{x_k^{t+1}} \log \tilde{p}_k^{t+1}$    `// conditional score approx.`

7        $x_k^t \sim \tilde{p}_\theta(\cdot \mid x_k^{t+1}, y) := \mathcal{N}\left(x_k^{t+1} + \sigma^2 \tilde{s}_k, \tilde{\sigma}^2\right)$    `// proposal`

8        $\tilde{p}_k^t \leftarrow \tilde{p}_\theta(y \mid x_k^t)$    `// twisting function (in eqs. (8), (13), or (14))`

9        $w_k \leftarrow p_\theta(x_k^t \mid x_k^{t+1}) \tilde{p}_k^t / [\tilde{p}_\theta(x_k^t \mid x_k^{t+1}, y) \tilde{p}_k^{t+1}]$    `// weight`

---

with the desired marginal, $p_\theta(x^0 \mid y)$. Comparing the diffusion conditional of eq. (5) to the SMC target of eq. (4) suggests SMC can be used.

For example, consider a first attempt at an SMC algorithm. Set the proposals as $r_T(x^T) = p(x^T)$ and $r_t(x^t \mid x^{t+1}) = p_\theta(x^t \mid x^{t+1})$ for $1 \le t \le T$, and weighting functions as $w_T(x^T) = w_t(x^t, x^{t+1}) = 1$ for $1 \le t \le T$ and $w_0(x^0, x^1) = p_{y|x^0}(y|x^0)$.

Substituting these choices into eq. (4) results in the desired final target $\nu_0 = p_\theta(x^{0:T} \mid y)$ with normalizing constant $\mathcal{L}_0 = p_\theta(y)$. As a result, the associated SMC algorithm produces a final set of $K$ samples and weights $\{x_k^0; w_k^0\}_{k=1}^K$ that provides an asymptotically accurate approximation $\mathbb{P}_K(x^0 \mid y) := (\sum_k^K w_k^0)^{-1} \sum_{k=1}^K w_k^0 \delta_{x_k^0}(x^0)$ to the desired $p_\theta(x^0 \mid y)$.

The approach above is simply importance sampling with proposal $p_\theta(x^{0:T})$; with all intermediate weights set to 1, one can skip resampling steps to reduce the variance of the procedure. Consequently, this approach will be impractical if $p_\theta(x^0 \mid y)$ is too dissimilar from $p_\theta(x^0)$ as only a small fraction of unconditional samples will have high likelihood: the number of particles required for accurate estimation of $p_\theta(x^{0:T} \mid y)$ is exponential in KL $\left[p_\theta(x^{0:T} \mid y) \parallel p_\theta(x^{0:T})\right]$ [3].

### 3.2 Twisted diffusion sampler

*Twisting* is a technique in the SMC literature intended to reduce the number of particles required for good approximation accuracy [15, 16]. Loosely, it introduces a sequence of *twisting functions* that modify the naive proposals and weighting functions, so that the resulting intermediate targets are closer to the final target of interests. We refer the reader to Naesseth et al. [25, Chapter 3.2] for background on twisting in SMC.

**Optimal twisting.** Consider defining the twisted proposals $r_t^*$ by multiplying the naive proposals $r_t$ described in Section 3.1 by $p_\theta(y \mid x^t)$ as

$$r_T^*(x^T) \propto p(x^T)p_\theta(y|x^T) \text{ and } r_t^*(x^t|x^{t+1}) \propto p_\theta(x^t|x^{t+1})p_\theta(y|x^t) \quad \text{for} \quad 0 \le t < T. \quad (6)$$

The factors $p_\theta(y \mid x^t)$ are the *optimal* twisting functions because they permit an SMC sampler that draws exact samples from $p_\theta(x^{0:T} \mid y)$ even when run with a single particle.

To see that a single particle is an exact sample, by Bayes rule, the proposals in eq. (6) reduce to

$$r_T^*(x^T) = p_\theta(x^T \mid y) \text{ and } r_t^*(x^t \mid x^{t+1}) = p_\theta(x^t \mid x^{t+1}, y) \quad \text{for} \quad 0 \le t < T. \quad (7)$$

As a result, if one samples $x^T \sim r_T^*(x^T)$ and $x^t \sim r_t^*(x^t \mid x^{t+1})$ for $t = T - 1, \ldots, 0$, by (i) the law of total probability and (ii) the chain rule of probability, one obtains $x^0 \sim p_\theta(x^0 \mid y)$ as desired.

However, we cannot readily sample from each $r_t^*$ because $p_\theta(y \mid x^t)$ is not analytically tractable. The challenge is that $p_\theta(y \mid x^t) = \int p_{y|x^0}(y|x^0)p_\theta(x^0 \mid x^t)dx^0$ depends on $x_t$ through $p_\theta(x^0 \mid x^t)$. The latter in turn requires marginalizing out $x^1, \ldots, x^{t-1}$ from the joint density $p_\theta(x^{0:t-1} \mid x^t)$, whose form depends on $t$ calls to the neural network $\hat{x}_\theta$.

**Tractable twisting.** To avoid this intractability, we approximate the optimal twisting functions by

$$\tilde{p}_\theta(y \mid x^t) := p_{y|x^0}(y|\hat{x}_\theta(x^t)) \approx p_\theta(y \mid x^t), \tag{8}$$

which is the likelihood function evaluated at $\hat{x}_\theta(x^t)$, the denoising estimate of $x^0$ at step $t$ from the diffusion model. This tractable twisting function $\tilde{p}_\theta(y \mid x^t)$ is the key ingredient needed to define the Twisted Diffusion Sampler (TDS, Algorithm 1). We motivate and develop its components below.

The approximation in eq. (8) offers two favorable properties. First, $\tilde{p}_\theta(y \mid x^t)$ is computable because it depends on $x^t$ only though one call to $\hat{x}_\theta$, instead of an intractable integral over many calls as in the case of optimal twisting. Second, $\tilde{p}_\theta(y \mid x^t)$ becomes an increasingly accurate approximation of $p_\theta(y \mid x^t)$, as $t$ decreases and $p_\theta(x^0 \mid x^t)$ concentrates on $\mathbb{E}_{p_\theta}[x^0 \mid x^t]$, which $\hat{x}_\theta$ is fit to approximate; at $t = 0$, where we can choose $\hat{x}_\theta(x^0) = x^0$, we obtain $\tilde{p}_\theta(y \mid x^0) = p_{y|x^0}(y|x^0)$.

We next use eq. (8) to develop a sequence of twisted proposals $\tilde{p}_\theta(x^t \mid x^{t+1}, y)$, to approximate the optimal proposals $p_\theta(x^t \mid x^{t+1}, y)$ in eq. (7). Specifically, we define twisted proposals as

$$\tilde{p}_\theta(x^t \mid x^{t+1}, y) := \mathcal{N}\left(x^t; x^{t+1} + \sigma^2 s_\theta(x^{t+1}, y), \tilde{\sigma}^2\right), \tag{9}$$

$$\text{where} \quad s_\theta(x^{t+1}, y) := s_\theta(x^{t+1}) + \nabla_{x^{t+1}} \log \tilde{p}_\theta(y \mid x^{t+1}) \tag{10}$$

is an approximation of the conditional score, $s_\theta(x^{t+1}, y) \approx \nabla_{x^{t+1}} \log p_\theta(x^{t+1} \mid y)$, and $\tilde{\sigma}^2$ is the proposal variance. For simplicity one could choose $\tilde{\sigma}^2 = \sigma^2$ to match the variance of $p_\theta(x^t \mid x^{t+1})$.

Equation (9) builds on previous works (e.g. [33, 31]) that seek to approximate the reversal of a conditional forward process. The gradient in eq. (10) is computed by back-propagating through $\hat{x}_\theta(x^{t+1})$. We further discuss this technique, which has been used before [e.g. in 19, 32], in Section 4.

**Twisted targets and weighting functions.** Because $\tilde{p}_\theta(x^t \mid x^{t+1}, y)$ will not in general coincide with the optimal twisted proposal $p_\theta(x^t \mid x^{t+1})$, we must introduce non-trivial weighting functions to ensure the resulting SMC sampler converges to the desired final target. In particular, we define *twisted* weighting functions as

$$w_t(x^t, x^{t+1}) := \frac{p_\theta(x^t \mid x^{t+1})\tilde{p}_\theta(y \mid x^t)}{\tilde{p}_\theta(y \mid x^{t+1})\tilde{p}_\theta(x^t \mid x^{t+1}, y)}, \qquad t = 0, \ldots, T-1 \tag{11}$$

and $w_T(x^T) := \tilde{p}_\theta(y \mid x^T)$. The weighting functions in eq. (11) recover the optimal, constant weighting functions if all other approximations at play are exact.

These tractable twisted proposals and weighting functions define intermediate targets that gradually approach the final target $\nu_0 = p_\theta(x^{0:T} \mid y)$. Substituting into eq. (4) each $\tilde{p}_\theta(x^t \mid x^{t+1}, y)$ in eq. (9) for $r_t(x^t \mid x^{t+1})$ and $w_t(x^t, x^{t+1})$ in eq. (11) and then simplifying we obtain the intermediate targets

$$\nu_t(x^{0:T}) \propto p_\theta(x^{0:T} \mid y)\left[\frac{\tilde{p}_\theta(y \mid x^t)}{p_\theta(y \mid x^t)} \prod_{t'=0}^{t-1} \frac{\tilde{p}_\theta(x^{t'} \mid x^{t'+1}, y)}{p_\theta(x^{t'} \mid x^{t'+1}, y)}\right]. \tag{12}$$

The right-hand bracketed term in eq. (12) can be understood as the discrepancy of $\nu_t$ from the final target $\nu_0$ accumulated from step $t$ to 0 (see Appendix A.3 for a derivation). As $t$ approaches 0, $\tilde{p}_\theta(y \mid x^t)$ improves as an approximation of $p_\theta(y \mid x^t)$, and the $t$-term product inside the bracket consists of fewer terms – the latter accounts for the discrepancy between practical and optimal proposals. Finally, at $t = 0$, because $\tilde{p}_\theta(y \mid x^0) = p_\theta(y \mid x^0)$ by construction, eq. (12) reduces to $\nu_0(x^{0:T}) = p_\theta(x^{0:T} \mid y)$, as desired.

**The TDS algorithm and asymptotic exactness.** Together, the twisted proposals $\tilde{p}_\theta(x^t \mid x^{t+1}, y)$ and weighting functions $w_t(x^t, x^{t+1})$ lead to *Twisted Diffusion Sampler*, or TDS (Algorithm 1). While Algorithm 1 states multinomial resampling for simplicity, in practice other resampling strategies (e.g. systematic [5, Ch. 9]) may be used as well. Under additional conditions, TDS provides arbitrarily accurate estimates of $p_\theta(x^0 \mid y)$. Crucially, this guarantee does not rely on assumptions on the accuracy of the approximations used to derive the twisted proposals and weights. Appendix A provides the formal statement with complete conditions and proof.

**Theorem 1.** *(Informal) Let* $\mathbb{P}_K(x^0) = (\sum_{k'}^K w_{k'})^{-1} \sum_{k=1}^K w_k \delta_{x_k^0}(x^0)$ *denote the discrete measure defined by the particles and weights returned by Algorithm 1 with $K$ particles. Under regularity conditions on the twisted proposals and weighting functions, $\mathbb{P}_K(x^0)$ converges setwise to $p_\theta(x^0 \mid y)$ as $K$ approaches infinity.*

### 3.3 TDS for inpainting, additional degrees of freedom

The twisting functions $\tilde{p}_\theta(y \mid x^t) := p_{y|x^0}(y|\hat{x}_\theta(x^t))$ we introduced above are one convenient option, but are sensible only when $p_{y|x^0}(y|\hat{x}_\theta(x^t))$ is differentiable and strictly positive. We now show how alternative twisting functions lead to proposals and weighting functions that address inpainting problems and more flexible conditioning specifications. In these extensions, Algorithm 1 still applies with the new definitions of twisting functions. Appendix A provides additional details, including the adaptation of TDS to variance preserving diffusion models [18].

**Inpainting.** Consider the case that $x^0$ can be segmented into observed dimensions $\mathbf{M}$ and unobserved dimensions $\bar{\mathbf{M}}$ such that we may write $x^0 = [x^0_{\mathbf{M}}, x^0_{\bar{\mathbf{M}}}]$ and let $y = x^0_{\mathbf{M}}$, and take $p_{y|x^0}(y|x^0) = \delta_y(x^0_{\mathbf{M}})$. The goal, then, is to sample $p_\theta(x^0 \mid x^0_{\mathbf{M}} = y) = p_\theta(x^0_{\bar{\mathbf{M}}} \mid x^0_{\mathbf{M}})\delta_y(x^0_{\mathbf{M}})$. Here we define the twisting function for each $t > 0$ as

$$\tilde{p}_\theta(y \mid x^t, \mathbf{M}) := \mathcal{N}\left(y; \hat{x}_\theta(x^t)_{\mathbf{M}}, t\sigma^2\right), \tag{13}$$

and set twisted proposals and weights according to eqs. (9) and (11). The variance in eq. (13) is chosen as $t\sigma^2 = \mathrm{Var}_{p_\theta}[x^t \mid x^0]$ for simplicity; in general, choosing this variance to more closely match $\mathrm{Var}_{p_\theta}[y \mid x^t]$ may be preferable. For $t = 0$, we define the twisting function analogously with small positive variance, for example as $\tilde{p}_\theta(y \mid x^0) = \mathcal{N}(y; x^0_{\mathbf{M}}, \sigma^2)$. This choice of simplicity changes the final target slightly; alternatively, the final twisting function, proposal, and weights may be chosen to maintain asymptotic exactness (see Appendix A.4).

**Inpainting with degrees of freedom.** We next consider the case when we wish to condition on some observed dimensions, but have additional degrees of freedom. For example in the context of motif-scaffolding in protein design, we may wish to condition on a functional motif $y$ appearing *anywhere* in a protein structure, rather than having a pre-specified set of indices $\mathbf{M}$ in mind. To handle this situation, we (i) let $\mathcal{M}$ be a set of possible observed dimensions, (ii) express our ambivalence in which dimensions are observed as $y$ using randomness by placing a uniform prior on $p(\mathbf{M}) = 1/|\mathcal{M}|$ for each $\mathbf{M} \in \mathcal{M}$, and (iii) again embed this new variable into our joint model to define the degrees of freedom likelihood by $p(y \mid x^0) = \sum_{\mathbf{M} \in \mathcal{M}} p_\theta(\mathbf{M}, y \mid x^0) = |\mathcal{M}|^{-1} \sum_{\mathbf{M} \in \mathcal{M}} p(y \mid x^0, \mathbf{M})$. Accordingly, we approximate $p_\theta(y \mid x^t)$ with the twisting function

$$\tilde{p}_\theta(y \mid x^t) := |\mathcal{M}|^{-1} \sum_{\mathbf{M} \in \mathcal{M}} \tilde{p}_\theta(y \mid x^t, \mathbf{M}), \tag{14}$$

with each $\tilde{p}_\theta(y \mid x^t, \mathbf{M})$ defined as in eq. (13). Notably, eqs. (13) and (14) coincide when $|\mathcal{M}| = 1$.

The sum in eq. (14) may be computed efficiently because each term depends on $x^t$ only through the same denoising estimate $\hat{x}_\theta(x^t)$, which must be computed only once. Since computation of $\hat{x}_\theta(x^t)$ is the expensive step, the overall run-time is not significantly increased by using even large $|\mathcal{M}|$.

### 3.4 TDS on Riemannian manifolds

TDS extends to Riemannian diffusion models on with little modification. Riemannian diffusion models [8] are structured like in the Euclidean case, but with conditionals defined by tangent normal distributions parameterized with a score approximation followed by a manifold projection step (see e.g. [4, 8]). When we assume that (as, e.g., in [39]) the model is associated with a denoising network $\hat{x}_\theta$, twisting functions are also constructed analogously. For conditional tasks defined by likelihoods, $\tilde{p}_\theta(y \mid x^t)$ in eq. (8) applies. For inpainting (and by extension, degrees of freedom), we propose

$$\tilde{p}_\theta(y \mid x^t) = \mathcal{TN}_{\hat{x}_\theta(x^t)_{\mathbf{M}}}(y; 0, t\sigma^2), \tag{15}$$

where $\mathcal{TN}_{\hat{x}_\theta(x^t)_{\mathbf{M}}}(0, t\sigma^2)$ is a tangent normal distribution centered on $\hat{x}_\theta(x^t)_{\mathbf{M}}$. As in the Euclidean case, $\tilde{p}_\theta(x^t \mid x^{t+1}, y)$ is defined with conditional score approximation $s_\theta(x^t, y) = s_\theta(x^t) + \nabla_{x^t} \log \tilde{p}(y \mid x^t)$, which is computable by automatic differentiation. Appendix B provides details.

## 4 Related work

There has been much recent work on conditional generation using diffusion models. These prior works demand either task specific conditional training, or involve unqualified approximations and can suffer from poor performance in practice. We discuss additional related work in Appendix C.

**Gradient guidance.** We use *gradient guidance* to refer to a general approach that incorporates conditioning information with gradients through the neural network $\hat{x}_\theta(x^t)$. For example, in an

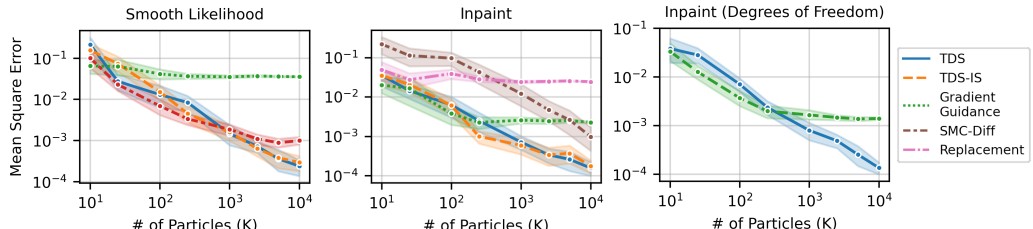

Figure 1: Errors of conditional mean estimations with 2 SEM error bars averaged over 25 replicates. TDS applies to all three tasks and provides increasing accuracy with more particles.

inpainting setting, Ho et al. [19] propose a Gaussian approximation $\mathcal{N}(y \mid \hat{x}_\theta(x^t)^{\mathbf{M}}, \alpha)$ to $p_\theta(y \mid x^t)$. This approximation motivates a modified transition distribution as in Equation (9), with the corresponding approximation to the conditional score as $s_\theta(x^t, y) = s_\theta(x^t) - \nabla_{x^t} \|y - \hat{x}_\theta(x^t)_M\|^2/\alpha$. This approach coincides exactly with TDS applied to the inpainting task when a single particle is used. Similar Gaussian approximations have been used by [6, 32] for other inverse problems.

Gradient guidance can also be used with non-Gaussian approximations; e.g. using $p_{y|x^0}(y|\hat{x}_\theta(x^t)) \approx p_\theta(y \mid x^t)$ for a given likelihood $p_{y|x^0}(y|x^0)$. This choice again recovers TDS with one particle.

Empirically, these heuristics can have unreliable performance, e.g. in image inpainting problems [40]. By comparison, TDS enjoys the benefits of gradient guidance by using it in proposals, while also providing a mechanism to eliminate approximation error by simulating additional particles.

**Replacement method.** The replacement method [33] is a method introduced for image inpainting using only unconditional diffusion models. The idea is to replace the observed dimensions of intermediate samples $x^t_{\mathbf{M}}$, with a noisy version of observation $x^0_{\mathbf{M}}$. However, it is a heuristic approximation and can lead to inconsistency between inpainted region and observed region [23]. Additionally, the replacement method applies only to inpainting problems. While recent work has extended the replacement method to linear inverse problems [e.g. 22], the approach provides no accuracy guarantees. It is unclear how to extend these methods to arbitrary differentiable likelihoods.

**SMC samplers for diffusion models.** Most closely related to the present work is SMC-Diff [34], which uses SMC to provide asymptotically accurate conditional samples for the inpainting problem. However, this prior work (i) is limited to the inpainting case, and (ii) provides asymptotic guarantees only under the assumption that the learned diffusion model exactly matches the forward noising process, which is rarely satisfied in practice. Also, SMC-Diff does not leverage twisting functions.

In concurrent work, Cardoso et al. [2] propose MCGdiff, an alternative SMC algorithm that uses the framework of auxiliary particle filtering [26] to provide asymptotically exact conditional inference. Compared to TDS, MCGdiff avoids computing gradients of the denoising network but applies only to linear inverse problems, with inpainting as a special case.

## 5 Simulation study and conditional image generation

We first test the dependence of the accuracy of TDS on the number of particles in synthetic settings with tractable exact conditionals in Section 5.1. Section 5.2 compares TDS to alternative approaches on class-conditional image generation, and an image inpainting experiment is included in Appendix D.2.2. See Appendix D for all additional details.

Our evaluation includes: (1) *TDS*; (2) *TDS-IS*, an importance sampler that uses TDS's proposal; (3) *IS*, a naive importance sampler described in Section 3.1; and (4) *Gradient Guidance*. For inpainting-type problems we further include: (5) *Replacement method*; and (6) *SMC-Diff*.

Each SMC sampler forms an approximation to $p_\theta(x^0 \mid y)$ with $K$ weighted particles $(\sum_{k'=1}^{K} w_{k'})^{-1} \cdot \sum_{k=1}^{K} w_k \delta_{x^0_k}$. Gradient guidance and replacement method are considered to form a similar particle-based approximation with $K$ independent samples viewed as $K$ particles with uniform weights.

**Compute Cost:** Compared to unconditional generation, TDS has compute cost that is (i) larger by a constant factor due to the need to backpropogate through the denoising network when computing the conditional score approximation and (ii) linear in the number of particles. As a result, the compute cost at inference cost is potentially large relative for accurate inference to be achieved.

By comparison, conditional training methods provide fast inference by *amortization* [13], in which most computation is done ahead of time. Hence they may be preferable for applications where sampling time computation is a primary concern. On the other hand, TDS may be preferable when the likelihood criterion is readily available (e.g. through an existing classifier on clean data) but training an amortized model poses challenges; for example, the labeled data, neural-network engineering expertise, and up-front compute resources required for amortization training can be prohibitive.

## 5.1 Applicability and precision of TDS in two dimensional simulations

We explore two questions in this section: (i) what sorts of conditioning information can be handled by TDS and other methods, and (ii) how does the precision of TDS depend on the number of particles?

To study these questions, we first consider an unconditional diffusion model $p_\theta(x^0)$ approximation of a bivariate Gaussian. For this choice, the marginals of the forward process are also Gaussian, and so we may define $p_\theta(x^{0:T})$ with an analytically tractable score function without neural network approximation. Consequently, we can analyze the performance without the influence of score network approximation errors. And the choice of a two-dimensional diffusion permits close approximation of exact conditional distributions by numerical integration that can then be used as ground truth.

We consider three test cases defining the conditional information: (1) Smooth likelihood: $y$ is an observation of the Euclidean norm of $x$ with Laplace noise, with $p_\theta(y \mid x^0) = \exp\{|\|x^0\|_2 - y|\}/2$. This likelihood is smooth almost everywhere.[3] (2) Inpainting: $y$ is an observation of the first dimension of $x^0$, with $p_\theta(y \mid x^0, \mathbf{M} = 0) = \delta_y(x_0^0)$. (3) Inpainting with degrees-of-freedom: $y$ is a an observation of either the first or second dimension of $x^0$, with $\mathcal{M} = \{0, 1\}$ and $p_\theta(y \mid x^0) = \frac{1}{2}[\delta_y(x_0^0) + \delta_y(x_1^0)]$.. In all cases we fix $y = 0$ and consider estimating $\mathbb{E}_{p_\theta}[x^0 \mid y]$.

Figure 1 reports the estimation error for the mean of the desired conditional distribution, i.e. $\|\sum w_k x_k^0 - \mathbb{E}_q[x^0 \mid y]\|_2$. TDS provides a computational-statistical trade-off: using more particles decreases mean square estimation error at the $O(1/K)$ parametric rate (note the slopes of $-1$ in log-log scale) as expected from standard SMC theory [5, Ch. 11]. This convergence rate is shared by TDS, TDS-IS, and IS in the smooth likelihood case, and by TDS, SMCDiff and in the inpainting case; TDS-IS, IS and SMCDiff are applicable however only in these respective cases, whereas TDS applicable in all cases. The only other method which applies to all three settings is Gradient Guidance, which exhibits significant estimation error and does not improve with many particles.

## 5.2 Class-conditional image generation

We next study the performance of TDS on diffusion models with neural network approximations to the score functions. In particular, we study the class-conditional image generation task, which involves sampling an image from $p_\theta(x^0 \mid y) \propto p_\theta(x^0)p_{y|x^0}(y|x^0)$, where $p_\theta(\cdot)$ is a pretrained diffusion model on images $x^0$, $y$ is a given image class, and $p_{y|x^0}(y|\cdot)$ is the classification likelihood. To assess the faithfulness of generation, we evaluate *classification accuracy* on predictions of conditional samples $x^0$ given $y$, made by the same classifier that specifies the likelihood. In all experiments, we follow the standard practice of returning the denoising mean on the final sample [18].

On the MNIST dataset, we compare TDS to TDS-IS, Gradient Guidance, and IS. Figure 2a compares the conditional samples of TDS and Gradient Guidance given class $y = 7$. Samples from Gradient Guidance have noticeable artifacts, and most of them do not resemble the digit 7; by contrast, TDS produces authentic and correct digits.

Figure 2b presents an ablation study of the effect of # of particles $K$ on MNIST. For all SMC samplers, more particles improve the classification accuracy, with $K = 64$ leading to nearly perfect accuracy. The performance of Gradient Guidance is constant with respect to $K$ (in expectation). Notably, for fixed $K$, TDS and TDS-IS have comparable performance and outperform Gradient guidance and IS.

We next apply TDS to higher dimension datasets. Figure 2c shows samples from TDS ($K = 16$) using a pre-trained diffusion model and a pretrained classifier on the ImageNet dataset ($256 \times 256 \times 3$ dimensions). These samples are qualitatively good and capture the class label. Appendix D.2.3 provides more samples, comparision to Classifier Guidance [9], and results for the CIFAR-10 dataset.

---

[3]This likelihood is smooth except at the point $x = (0, 0)$.

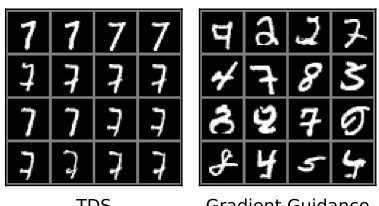 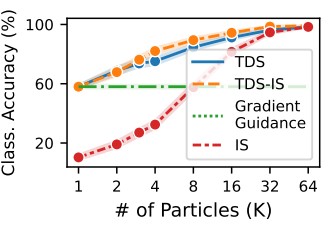 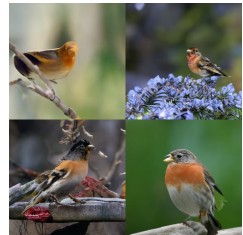

|      TDS      |  Gradient Guidance  |

(a) MNIST: samples by TDS ($K = 64$) and Gradient Guidance given class '7'. TDS samples are randomly selected out of 64 particles in a single SMC run.

(b) MNIST: classification accuracy vs. $K$. Results are averaged over 1,000 runs with error bars denoting 2 standard errors.

(c) ImageNet: samples by TDS ($K = 16$) given class 'brambling' from 4 independent SMC runs.

Figure 2: Image class-conditional generation task.

TDS can be extended by exponentiating twisting functions with a *twist scale*. This extension is related to the existing literature of Classifier Guidance that considers re-scaling the gradient of the log classification probability. See Appendix D.2.1 for details and ablation study.

## 6 Case study in computational protein design: the motif-scaffolding problem

The biochemical functions of proteins are typically imparted by a small number of atoms, known as a *motif*, that are stabilized by the overall protein structure, known as the *scaffold* [36]. A central task in protein design is to identify stabilizing scaffolds in response to motifs expected to confer function. We here describe an application of TDS to this task, and compare TDS to the state-of-the-art conditionally-trained model, RFdiffusion. See additional details in Appendix E.

Given a generative model supported on designable protein structures $p_\theta(x^0)$, suitable scaffolds may be constructed by solving a conditional generative modeling problem [34]. Complete structures are first segmented into a motif $x_\mathbf{M}^0$ and a scaffold $x_{\overline{\mathbf{M}}}^0$, i.e. $x^0 = [x_\mathbf{M}^0, x_{\overline{\mathbf{M}}}^0]$. Putative compatible scaffolds are then identified by (approximately) sampling from $p_\theta(x_{\overline{\mathbf{M}}}^0 \mid x_\mathbf{M}^0)$ [34].

While the conditional generative modeling approach to motif-scaffolding has produced functional, experimentally validated structures for certain motifs [37], the general problem remains open. Moreover, current methods for motif-scaffolding require one to specify the location of the motif within the primary sequence of the full scaffold; this choice can require expert knowledge and trial and error.

We hypothesized that improved motif-scaffolding could be achieved through accurate conditional sampling. To this end, we applied TDS to FrameDiff, a Riemannian diffusion model that parameterizes protein backbones as a collection of $N$ rigid bodies (known as residues) in the manifold $SE(3)^N$ [39].[4] Each of the $N$ elements of $SE(3)$ consists of a rotation matrix and a translation that parameterize the locations of the backbone atoms of each residue.

**Likelihood, twisting, and degrees of freedom.** The basis of our approach to the motif scaffolding problem is analogous to the inpainting case described in Section 3.3. We let $p_{y|x^0}(y|x^0) = \delta_y(x_\mathbf{M}^0)$, where $y \in SE(3)^M$ describes the coordinates of backbone atoms of an $M = |\mathcal{M}|$ residue motif. As such, to define twisting function we adopt the Riemannian TDS formulation described in Section 3.4.

To eliminate the requirement that the placement of the motif within the scaffold be pre-specified, we incorporate the motif placement as a degree of freedom. We (i) treat the indices of the motif within the chain as a mask $\mathbf{M}$, (ii) let $\mathcal{M}$ be a set of possible masks of size equal to the length of the motif, and (iii) apply Equation (14) to average over these possible masks.

For some scaffolding problems, it is known that all motif residues must appear with contiguous indices. In this case we choose $\mathcal{M}$ to be the set of all possible contiguous masks. However, when motif residues are only known to appear in two or more possibly discontiguous blocks, the number of possible placements can be too large and we choose $\mathcal{M}$ by randomly sampling at most some maximum number (# Motif Locs.) of masks.

We similarly eliminate the global translation and rotation of the motif as a degree of freedom. The motivation is that restricting to one possible pose of the motif narrows the conditional distribution,

---

[4]https://github.com/jasonkyuyim/se3_diffusion

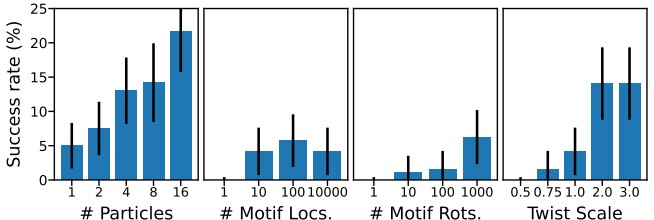

| Scaffold size | TDS & FrameDiff | RF diffusion |
|---|---|---|
| <100 res. | 9 | 3 |
| $\geq$ 100 res. | 2 | 8 |
| Overall | 11 | 11 |

(a) TDS motif-scaffolding success rate (test case `5IUS`) improves with more particles, and degrees of freedom, and twist-scale.

(b) # problems with higher success rate

Figure 3: Protein motif-scaffolding case study results

thereby making inference more challenging. For translation, we use a likelihood that is invariant to the motif's center-of-mass and placement in the final scaffold by choosing $p_{y|x^0}(y|x) = \delta_{Py}(Px_{\mathbf{M}})$ where $P$ is a projection matrix that removes the center of mass of a vector [see e.g. 39, section 3.3]. For rotation, we average the likelihood across some number (`# Motif Rots.`) of possible rotations.

**Ablation study.** We first examine the impact of several parameters of TDS on success rate in an in silico *self-consistency* evaluation [34]. We begin with single problem (`5IUS`) in the benchmark set introduced by [37], before testing on the full set. Figure 3a (Left) shows that success rate increases monotonically with the number of particles. Non-zero success rates in this setting required accounting for multiple motif locations; Figure 3a (Left) uses 1,000 possible motif locations and 100 rotations.

We tested the impact of the degrees of freedom by evaluating the success rate of TDS (K=1) with increasing motif locations and 100 rotations (Figure 3a Center Left), and increasing rotations and 1,000 locations (Figure 3a Center Right). The success rate was 0% without accounting for either degree of freedom, and increased with larger numbers of locations and rotations. We also explored including a heuristic twist scale as considered for image tasks (Section 5.2); in this case, the twist scale is a multiplicative factor on the logarithm of the twisting function. Figure 3a (Right) shows larger twist scales gave higher success rates on this test case, where we use 8 particles, 1,000 possible motif locations and 100 rotations. However, this trend is not monotonic for all problems (Figure P).

**Evaluation on full benchmark.** We next evaluate on TDS on a benchmark set of 24 motif-scaffolding problems [37] and compare to the previous state of the art, RFdiffusion. RFdiffusion operates on the same rigid body representation of protein backbones as FrameDiff. TDS is run with K=8, twist scale=2, and 100 rotations and 1,000 motif location degrees of freedom (100,000 combinations total).

Overall, TDS (applied to FrameDiff) and RFdiffusion have comparable performance (Figure 3b); each provides a success rate higher than the other in 11/24 cases; on two problems both methods have a 0% success rate (full results in Figure O). This performance is obtained despite the fact that FrameDiff, unlike RFdiffusion, is not trained to perform motif scaffolding. The division between problems on which each method performs well is primarily explained by total scaffold length, with TDS providing higher success rates on smaller scaffolds.

We suspect the shift in performance with scaffold length owes properties of the underlying diffusion models. First, long backbones generated unconditionally by RFdiffusion are designable with higher frequency than those generated by FrameDiff [39]. Second, unlike RFdiffusion, FrameDiff can not condition on the fixed motif sequence.

# 7 Discussion

We propose TDS, a practical and asymptotically exact conditional sampling algorithm for diffusion models. We compare TDS to other approaches and demonstrate the effectiveness and flexibility of TDS on image class conditional generation and inpainting tasks. On protein motif-scaffolding problems with short scaffolds, TDS outperforms the current (conditionally trained) state of the art.

A limitation of TDS is its requirement for additional computes to simulate multiple particles. While we observe improved performance with just two particles in some cases, the optimal number is problem dependent. Moreover, the computational efficiency depends on how closely the twisting functions approximate exact conditionals, which depends on the unconditional model and conditioning information. Lastly, choosing twisting functions for generic constraints may be challenging. Our future work will focus on addressing these limitations and improving the computational efficiency.

## Acknowledgements

We thank Hai-Dang Dau, Arnaud Doucet, Joe Watson, and David Juergens for helpful discussion, Jason Yim for discussions and code, and David Baker for additional guidance.

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

# Appendix

## A  Twisted Diffusion Sampler additional details

In this section we provide additional details on TDS. Appendix A.1 describes its generalization to alternative common formulations of diffusion models. Appendix A.2 describes the choices of proposal variance and resampling strategy. Appendix A.3 provides a derivation of Equation (12). Appendix A.4 describes modifications to the final step of TDS in inpainting and inpainting with degrees of freedom applications. And Appendix A.5 provides a full statement and proof of Theorem 1.

### A.1  Diffusion models with non-constant noise variance and variance-preserving scaling

TDS as developed in Section 3 assumed noise with constant variance $\sigma^2$ added at each timestep. This choice corresponds to a particular discretization of a "variance exploding" (VE) diffusion model [33]. Here we describe generalizations of TDS to non-constant variance schedules, and to variance preserving (VP) diffusion models [18]. We adopt the VP formulation in our experiments; though it introduces additional notation, it is known to work well in practice [33].

**TDS algorithm for variance exploding diffusion models.**    VE diffusion models define the forward process $q$ by

$$q(x^{1:T} \mid x^0) := \prod_{t=1}^{T} q(x^t \mid x^{t-1}), \qquad q(x^t \mid x^{t-1}) := \mathcal{N}(x^t; x^{t-1}, \sigma_t^2). \qquad (16)$$

where $\sigma_t^2$ is an increasing sequence of variances such that $q(x^T) \approx \mathcal{N}(0, \bar{\sigma}_T^2)$, with $\bar{\sigma}_t^2 := \sum_{t'=1}^{t} \sigma_{t'}^2$ for $t = 1, \cdots, T$. And so one can set $p(x^T) = \mathcal{N}(0, \bar{\sigma}_T^2)$ to match $q(x^T)$. Notably, Equation (16) implies the conditional $q(x^t \mid x^0) = \mathcal{N}(x^t; x^0, \bar{\sigma}_t^2)$.

The reverse diffusion process $p_\theta$ is parameterized as

$$p_\theta(x^{0:T}) := p(x^T) \prod_{t=T}^{1} p_\theta(x^{t-1} \mid x^t), \qquad p_\theta(x^{t-1} \mid x^t) := \mathcal{N}(x^{t-1}; x^t + \sigma_t^2 s_\theta(x^t, t), \sigma_t^2)$$

where the score network $s_\theta$ is modeled through a denoiser network $\hat{x}_\theta$ by $s_\theta(x^t, t) := (\hat{x}_\theta(x^t, t; \theta) - x^t)/\bar{\sigma}_t^2$. Note that the constant schedule is a special case where $\sigma_t^2 = \sigma^2$, and $\bar{\sigma}_t^2 = t\sigma^2$ for all $t$.

The TDS algorithm for general VE models is described in Algorithm 1, where $t\sigma^2$ in Line 6 is replaced by $\bar{\sigma}_t^2$ and $\sigma^2$ in Line 7 is replaced by $\sigma_t^2$.

**Extension to variance preserving diffusion models.**    Another widely used diffusion framework is variance preserving (VP) diffusion models [18]. VP models define the forward process $q$ by

$$q(x^{1:T} \mid x^0) := \prod_{t=1}^{T} q(x^t \mid x^{t-1}), \qquad q(x^t \mid x^{t-1}) := \mathcal{N}(x^t; \sqrt{1 - \sigma_t^2} x^{t-1}, \sigma_t^2)$$

where $\sigma_t^2$ is a sequence of increasing variances chosen such that $q(x^T) \approx \mathcal{N}(0, 1)$, and so one can set $p(x^T) = \mathcal{N}(0, 1)$. Define $\alpha_t := 1 - \sigma_t^2$, $\bar{\alpha}_t := \prod_{t'=1}^{t} \alpha_{t'}$, and $\bar{\sigma}_t^2 := 1 - \bar{\alpha}_t$. Then the marginal conditional of eq. (16) is $q(x^t \mid x^0) = \mathcal{N}(x^t; \sqrt{\bar{\alpha}_t} x^0, \bar{\sigma}_t^2)$. The reverse diffusion process $p_\theta$ is parameterized as

$$p_\theta(x^{0:T}) := p(x^T) \prod_{t=T}^{1} p_\theta(x^{t-1} \mid x^t), \quad p_\theta(x^{t-1} \mid x^t) := \mathcal{N}(x^{t-1}; \frac{1}{\sqrt{\alpha_t}} x^t + \frac{\sigma_t^2}{\sqrt{\alpha_t}} s_\theta(x^t, t), \sigma_t^2)$$

where $s_\theta$ is now defined through the denoiser $\hat{x}_\theta$ by $s_\theta(x^t, t) := (\sqrt{\bar{\alpha}_t}\hat{x}_\theta(x^t, t; \theta) - x^t)/\bar{\sigma}_t^2$.

TDS in Algorithm 1 extends to VP models as well, where the conditional score approximation in Line 6 is changed to $\tilde{s}_k \leftarrow (\sqrt{\bar{\alpha}_{t+1}}\hat{x}_\theta(x_k^{t+1}) - x_k^{t+1})/\bar{\sigma}_{t+1}^2 + \nabla_{x_k^{t+1}} \log \tilde{p}_k^{t+1}$, and the proposal in Line 7 is changed to $x_k^t \sim \tilde{p}_\theta(\cdot \mid x_k^{t+1}, y) := \mathcal{N}\left(\frac{1}{\sqrt{\alpha_{t+1}}}x_k^{t+1} + \frac{\sigma_{t+1}^2}{\sqrt{\alpha_{t+1}}}\tilde{s}_k, \tilde{\sigma}^2\right)$.

## A.2 TDS parameters

**Proposal variance.** The proposal distribution in Line 7 of Algorithm 1 is associated with a variance parameter $\tilde{\sigma}^2$. In general, this parameter can be dependent on the time step, i.e. replacing $\tilde{\sigma}^2$ by some $\tilde{\sigma}_{t+1}^2$ in Line 7. Unless otherwise specified, we set $\tilde{\sigma}_{t+1}^2 := \mathrm{Var}_{p_\theta}[x^t \mid x^{t+1}]$ the variance of the unconditional diffusion model (typically learned along with the score network during training).

**Resampling strategy.** The mulinomial resampling strategy in Line 4 of Algorithm 1 can be replaced by other strategies, see [25, Chapter 2] for an overview. In our experiments, we use the systematic resampling strategy.

In addition, one can consider setting an effective sample size (ESS) threshold (between 0 and 1), and only when the ESS is smaller than this threshold, the resampling step is triggered. ESS thresholds for resampling are commonly used to improve efficiency of SMC algorithms [see e.g. 25, Chapter 2.2.2], but for simplicity we use TDS with resampling at every step unless otherwise specified.

## A.3 Derivation of Equation (12)

Equation (12) illustrated how the extended intermediate targets $\nu_t(x^{0:T})$ provide approximations to the final target $p_\theta(x^{0:T} \mid y)$ that become increasingly accurate as $t$ approaches 0. We obtain Equation (12) by substituting the proposal distributions in Equation (9) and weights in Equation (11) into Equation (4) and simplifying.

$$\nu_t(x^{0:T}) \propto \left[p(x^T)\prod_{t'=0}^{T-1}\tilde{p}_\theta(x^{t'} \mid x^{t'+1}, y)\right]\left[\tilde{p}(y \mid x^T)\prod_{t'=t}^{T-1}\frac{p_\theta(x^{t'} \mid x^{t'+1})\tilde{p}_\theta(y \mid x^{t'})}{\tilde{p}_\theta(y \mid x^{t'+1})\tilde{p}_\theta(x^{t'} \mid x^{t'+1}, y)}\right]$$

Rearrange and cancel $p_\theta$ and $\tilde{p}_\theta$ terms.

$$= \left[p(x^T)\prod_{t'=t}^{T-1}p_\theta(x^{t'} \mid x^{t'+1})\right]\left[\tilde{p}_\theta(y \mid x^t)\prod_{t'=0}^{t-1}\tilde{p}_\theta(x^{t'} \mid x^{t'+1}, y)\right]$$

Group $p_\theta$ terms by chain rule of probability.

$$= p_\theta(x^{t:T})\tilde{p}_\theta(y \mid x^t)\prod_{t'=0}^{t-1}\tilde{p}_\theta(x^{t'} \mid x^{t'+1}, y)$$

Apply Bayes' rule and note that $p_\theta(y \mid x^{t:T}) = p_\theta(y \mid x^t)$.

$$\propto p_\theta(x^{t:T} \mid y)\frac{\tilde{p}_\theta(y \mid x^t)}{p(y \mid x^t)}\prod_{t'=0}^{t-1}\tilde{p}_\theta(x^{t'} \mid x^{t'+1}, y)$$

Note that $p_\theta(x^{0:T} \mid y) = p_\theta(x^{t:T} \mid y)\prod_{t=0}^{t'-1}p_\theta(x^{t'} \mid x^{t'+1}, y)$.

$$= p_\theta(x^{0:T} \mid y)\left[\frac{\tilde{p}_\theta(y \mid x^t)}{p(y \mid x^t)}\prod_{t'=0}^{t-1}\frac{\tilde{p}_\theta(x^{t'} \mid x^{t'+1}, y)}{p_\theta(x^{t'} \mid x^{t'+1}, y)}\right].$$

The final line is the desired expression in Equation (12).

## A.4 Inpainting and degrees of freedom final steps for asymptotically exact target

The final ($t = 0$) twisting function for inpainting and inpainting with degrees of freedom described in Section 3.3 do not satisfy the assumption of Theorem 2 that $\tilde{p}_\theta(y \mid x^0) = p_{y|x^0}(y|x^0)$. This choice introduces error in the final target of TDS relative to the exact conditional $p_\theta(x^{0:T} \mid y)$.

For inpainting, to maintain asymptotic exactness one may instead choose the final proposal and weights as

$$\tilde{p}_\theta(x^0 \mid x^1, y; \mathbf{M}) := \delta_y(x_{\mathbf{M}}^0) p_\theta(x_{\overline{\mathbf{M}}}^0 \mid x^1) \quad \text{and} \quad w_0(x^0, x^1) = 1.$$

One can verify the resulting final target is $p_\theta(x^0 \mid x_{\mathbf{M}}^0 = y)$ according to eq. (4).

Similarly, for inpainting with degrees of freedom, one may define the final proposal and weight as

$$\tilde{p}_\theta(x^0 \mid x^1, y, \mathcal{M}) := \sum_{\mathbf{M} \in \mathcal{M}} \frac{p_\theta(x_{\mathbf{M}}^0 \mid x^1, \mathbf{M})}{\sum_{\mathbf{M}' \in \mathcal{M}} p_\theta(x_{\mathbf{M}'}^0 \mid x^1, \mathbf{M}')} \delta_y(x_{\mathbf{M}}^0) p_\theta(x_{\overline{\mathbf{M}}}^0 \mid x^1) \quad \text{and} \quad w_0(x^0, x^1) = 1.$$

### A.5 Asymptotic accuracy of TDS – additional details and full theorem statement

In this section we (i) characterize sufficient conditions on the model and twisting functions under which TDS provides arbitrarily accurate estimates as the number of particles is increased and (ii) discuss when these conditions will hold in practice for the twisting function $\tilde{p}_\theta(y \mid x^t)$ introduced in Section 3.

We begin with a theorem providing sufficient conditions for asymptotic accuracy of TDS.

**Theorem 2.** *Let $p_\theta(x^{0:T})$ be a diffusion generative model (defined by eqs. (1) and (3)) with*

$$p_\theta(x^t \mid x^{t+1}) = \mathcal{N}\left(x^t \mid x^{t+1} + \sigma_{t+1}^2 s_\theta(x^{t+1}), \sigma_{t+1}^2 I\right),$$

*with variances $\sigma_1^2, \ldots, \sigma_T^2$. Let $\tilde{p}_\theta(y \mid x^t)$ be twisting functions, and*

$$r_t(x^t \mid x^{t+1}) = \mathcal{N}\left(x^t \mid x^{t+1} + \sigma_{t+1}^2[s_\theta(x^{t+1}) + \nabla_{x^{t+1}} \log \tilde{p}_\theta(y \mid x^{t+1})], \tilde{\sigma}_{t+1}^2\right)$$

*be proposals distributions for $t = 0, \ldots, T - 1$, and let $\mathbb{P}_K = \sum_{k=1}^K w_k^0 \delta_{x_k^0}$ for weighted particles $\{(x_k^0, w_k^0)\}_{k=1}^K$ returned by Algorithm 1 with $K$ particles. Assume*

(a) *the final twisting function is the likelihood, $\tilde{p}_\theta(y \mid x^0) = p_{y|x^0}(y|x^0)$,*

(b) *the first twisting function $\tilde{p}_\theta(y \mid x^T)$, and the ratios of subsequent twisting functions $\tilde{p}_\theta(y \mid x^t)/\tilde{p}_\theta(y \mid x^{t+1})$ are positive and bounded,*

(c) *each $\log \tilde{p}_\theta(y \mid x^t)$ with $t > 0$ is continuous and has bounded gradients in $x^t$, and*

(d) *the proposal variances are larger than the model variances, i.e. for each $t$, $\tilde{\sigma}_t^2 > \sigma_t^2$.*

*Then $\mathbb{P}_K$ converges setwise to $p_\theta(x^0 \mid y)$ with probability one, that is for every set $A$, $\lim_{K \to \infty} \mathbb{P}_K(A) = \int_A p_\theta(x^0 \mid y) dx^0$.*

The assumptions of Theorem 2 are readily satisfied in common applications.

- Assumption (a) may be satisfied by construction by choosing $\tilde{p}_\theta(y \mid x^0) = p_{y|x^0}(y|x^0)$ by defining $\hat{x}_\theta(x^0, t = 0) = x^0$.

- Assumption (b) is satisfied if (i) $p_{y|x^0}(y|x)$ is smooth in $x$ and everywhere positive and (ii) $\tilde{p}_\theta(y \mid x^t) = p_{y|x^0}(y|\hat{x}_\theta(x^t))$ where $\hat{x}_\theta(x^t)$ takes values in some compact domain. An alternative sufficient condition for Assumption (b) is for $p_{y|x^0}(y|x)$ to be positive and bounded away from zero; this latter condition will hold when, for example, $p_{y|x^0}(y|x)$ is a classifier fit with regularization.

- Assumption (c) is the strongest assumption. It will be satisfied, for example, if (i) $\tilde{p}_\theta(y \mid x^t) = p_{y|x^0}(y|\hat{x}_\theta(x^t))$, and (ii) $p_{y|x^0}(y|x)$ and $\hat{x}_\theta(x^t)$ are smooth in $x$ and $x^t$, with uniformly bounded gradients. While smoothness of $\hat{x}_\theta(\cdot, t)$ can be encouraged by the use of skip-connections and regularization, $\hat{x}_\theta(\cdot, t)$ may present sharp transitions, particularly for $t$ close to zero.

- Assumption (d), that the proposal variances satisfy $\tilde{\sigma}_t^2 > \sigma_t^2$ is likely not needed for the result to hold. However, this assumption permits usage of existing SMC theoretical results in the proof; in practice, our experiments use $\tilde{\sigma}_t^2 = \sigma_t^2$, but alternatively the assumption could be met by inflating each $\tilde{\sigma}_t^2$ by some arbitrarily small $\delta$ without markedly impacting the behavior of the sampler.

**Proof of Theorem 2:** Theorem 2 characterizes a set of conditions under which SMC algorithms converge. We restate this result below in our own notation.

**Theorem 3** (Chopin and Papaspiliopoulos [5] – Proposition 11.4). *Let $\{(x_k^0, w_k^0)\}_{k=1}^K$ be the particles and weights returned at the last iteration of a sequential Monte Carlo algorithm with $K$ particles using multinomial resampling. If each weighting function $w_t(x^t, x^{t+1})$ is positive and bounded, then for every bounded, $\nu_0$-measurable function $\phi$ of $x^t$*

$$\lim_{K\to\infty} \sum_{k=1}^K w_k^0 \phi(x_k^0) = \int \phi(x^0)\nu_0(x^0)dx^0.$$

*with probability one.*

An immediate consequence of Theorem 3 is the setwise convergence of the discrete measures, $\hat{P}_K = \sum_{k=1}^K w_k^0 \delta_{x_k^0}$. This can be seen by taking for each $\phi(x) = \mathbb{I}[x \in A]$ for any $\nu_0$-measurable set $A$. The theorem applies both in the Euclidean setting, where each $x_k^t \in \mathbb{R}^D$, as well as the Riemannian setting.

We now proceed to prove Theorem 2.

*Proof.* To prove the theorem we show (i) the $x^0$ marginal final target $\nu_0$ is $p_\theta(x^0 \mid y)$ and then (ii) $\mathbb{P}_K$ converges setwise to $\nu_0$.

We first show (i) by manipulating $\nu_0$ in Equation (4) to obtain $p_\theta(x^{0:T} \mid y)$. From Equation (4) we first have

$$\nu_0(x^{0:T}) = \frac{1}{\mathcal{L}_0}\left[r(x^T)\prod_{t=0}^{T-1}r_t(x^t \mid x^{t+1})\right]\left[w_T(x^T)\prod_{t=0}^{T-1}w_{t'}(x^t, x^{t+1})\right]$$

Substitute in weights from eq. (11).

$$= \frac{1}{\mathcal{L}_0}\left[p(x^T)\prod_{t=0}^{T-1}r_t(x^t \mid x^{t+1})\right]\left[\tilde{p}_\theta(y \mid x^T)\prod_{t=0}^{T-1}\frac{p_\theta(x^t \mid x^{t+1})\tilde{p}_\theta(y \mid x^t)}{\tilde{p}_\theta(y \mid x^{t+1})r_t(x^t \mid x^{t+1})}\right]$$

Rearrange $p_\theta$ and $r_t$ terms, and cancel out $r_t$ terms.

$$= \frac{1}{\mathcal{L}_0}\left[p(x^T)\prod_{t=0}^{T-1}p_\theta(x^t \mid x^{t+1})\right]\left[\tilde{p}_\theta(y \mid x^T)\prod_{t=0}^{T-1}\frac{\cancel{r_t(x^t \mid x^{t+1})}\tilde{p}_\theta(y \mid x^t)}{\tilde{p}_\theta(y \mid x^{t+1})\cancel{r_t(x^t \mid x^{t+1})}}\right]$$

Collapse $p_\theta$ terms.

$$= \frac{1}{\mathcal{L}_0}p_\theta(x^{0:T})\left[\prod_{t=0}^{T-1}\frac{\tilde{p}_\theta(y \mid x^t)}{\tilde{p}_\theta(y \mid x^{t+1})}\right]\tilde{p}_\theta(y \mid x^T)$$

Cancel out $\tilde{p}_\theta$ terms.

$$= \frac{1}{\mathcal{L}_0}p_\theta(x^{0:T})\tilde{p}_\theta(y \mid x^0)$$

Recognize $\tilde{p}_\theta(y \mid x^0) = p_{y|x^0}(y|x^0)$ by Assm.(a) and apply Bayes' rule with $\mathcal{L}_0 = p_\theta(y)$.

$$= p_\theta(x^{0:T} \mid y).$$

The final line reveals that once we marginalize out $x^{1:T}$ we obtain $\nu_0(x^0) = p(x^0 \mid y)$ as desired.

We next show that $\mathbb{P}_K$ converges to $\nu_0$ with probability one by applying Theorem 3. To apply Theorem 3 it is sufficient to show that the weights at each step are upper bounded, as they are defined through (ratios of) probabilities and hence are positive. Since there are a finite number of steps $T$, it is enough to show that each $w_t$ is bounded. The inital weight is the initial twisting function, $w_T(x^T) = \tilde{p}_\theta(y \mid x^T)$, which is bounded by Assumption (b). So we proceed to intermediate weights.

To show that the weighting functions at subsequent steps are bounded, we decompose the log-weighting functions as

$$\log w_t(x^t, x^{t+1}) = \log\frac{\tilde{p}_\theta(y \mid x^t)}{\tilde{p}_\theta(y \mid x^{t+1})} + \log\frac{p_\theta(x^t \mid x^{t+1})}{r_t(x^t \mid x^{t+1})},$$

and show independently that $\log \tilde{p}_\theta(y \mid x^t)/\tilde{p}_\theta(y \mid x^{t+1})$ and $\log p_\theta(x^t \mid x^{t+1})/r_t(x^t \mid x^{t+1})$ are bounded. The first term $\log \tilde{p}_\theta(y \mid x^t)/\tilde{p}_\theta(y \mid x^{t+1})$ is again bounded by Assumption (b), and we proceed to the second.

That $\log p_\theta(x^t \mid x^{t+1})/r_t(x^t \mid x^{t+1})$ is bounded follows from Assumptions (c) and (d). First write

$$p_\theta(x^t \mid x^{t+1}) = \mathcal{N}\left(x^t \mid \hat{\mu}, \sigma_{t+1}^2 I\right)$$

with $\hat{\mu} = x^{t+1} + \sigma_{t+1}^2 s_\theta(x^{t+1})$, and

$$r_t(x^t \mid x^{t+1}) = \mathcal{N}\left(x^t \mid \hat{\mu}_\psi, \tilde{\sigma}_{t+1}^2 I\right),$$

for $\hat{\mu}_\psi = \hat{\mu} + \sigma_{t+1}^2 \nabla_{x^{t+1}} \log \tilde{p}_\theta(y \mid x^{t+1})$. The log-ratio then simplifies as

$$\log \frac{p_\theta(x^t \mid x^{t+1})}{r_t(x^t \mid x^{t+1})} = \log \frac{|2\pi\sigma_{t+1}^2 I|^{-1/2} \exp\{-(2\sigma_{t+1}^2)^{-1}\|\hat{\mu} - x^t\|^2\}}{|2\pi\tilde{\sigma}_{t+1}^2 I|^{-1/2} \exp\{-(2\tilde{\sigma}_{t+1}^2)^{-1}\|\hat{\mu}_\psi - x^t\|^2\}}$$

Rearrange and let $C = \log|2\pi\sigma_{t+1}^2 I|^{-1/2}/|2\pi\tilde{\sigma}_{t+1}^2 I|^{-1/2}$

$$= \frac{-1}{2}\left[\sigma_{t+1}^{-2}\|\hat{\mu} - x^t\|^2 - \tilde{\sigma}_{t+1}^{-2}\|\hat{\mu}_\psi - x^t\|^2\right] + C$$

Expand and rearrange $\|\hat{\mu}_\psi - x^t\|^2 = \|\hat{\mu} - x^t\|^2 + 2\langle\hat{\mu}_\psi - \hat{\mu}, \hat{\mu} - x^t\rangle + \|\hat{\mu}_\psi - \hat{\mu}\|^2$

$$= \frac{-1}{2}\left[(\sigma_{t+1}^{-2} - \tilde{\sigma}_{t+1}^{-2})\|\hat{\mu} - x^t\|^2 - 2\tilde{\sigma}_{t+1}^{-2}\langle\hat{\mu}_\psi - \hat{\mu}, \hat{\mu} - x^t\rangle - \tilde{\sigma}_{t+1}^2\|\hat{\mu}_\psi - \hat{\mu}\|^2\right] + C$$

Let $C' = C - \frac{1}{2}\tilde{\sigma}_{t+1}^2\|\hat{\mu}_\psi - \hat{\mu}\|^2$ and rearrange. Note that $\|\hat{\mu}_\psi - \hat{\mu}\|^2 < \infty$ by Assm. (c).

$$= \frac{-1}{2}(\sigma_{t+1}^{-2} - \tilde{\sigma}_{t+1}^{-2})\|\hat{\mu} - x^t\|^2 + \tilde{\sigma}_{t+1}^{-2}\langle\hat{\mu}_\psi - \hat{\mu}, \hat{\mu} - x^t\rangle + C'$$

Apply Cauchy-Schwarz

$$\leq \frac{-1}{2}(\sigma_{t+1}^{-2} - \tilde{\sigma}_{t+1}^{-2})\|\hat{\mu} - x^t\|^2 + \tilde{\sigma}_{t+1}^{-2}\|\hat{\mu}_\psi - \hat{\mu}\| \cdot \|\hat{\mu} - x^t\| + C'$$

Upper-bounding using that $\max_x \frac{-a}{2}x^2 + bx = \frac{b^2}{2a}$ for $a = \sigma_{t+1}^{-2} - \tilde{\sigma}_{t+1}^{-2} > 0$ by Assm. (d).

$$\leq \frac{1}{2}\frac{(\tilde{\sigma}_{t+1}^{-2}\|\hat{\mu}_\psi - \hat{\mu}\|)^2}{\sigma_{t+1}^{-2} - \tilde{\sigma}_{t+1}^{-2}} + C'$$

$$= \frac{\tilde{\sigma}_{t+1}^{-4}}{2(\sigma_{t+1}^{-2} - \tilde{\sigma}_{t+1}^{-2})}\|\hat{\mu}_\psi - \hat{\mu}\|^2 + C'$$

Note that $\hat{\mu}_\psi = \hat{\mu} + \sigma_{t+1}^2 \nabla_{x^{t+1}} \log \tilde{p}_\theta(y \mid x^{t+1})$.

$$= \frac{\tilde{\sigma}_{t+1}^{-4}}{2(\sigma_{t+1}^{-2} - \tilde{\sigma}_{t+1}^{-2})}\sigma_{t+1}^4\|\nabla_{x^{t+1}} \log \tilde{p}_\theta(y \mid x^{t+1})\|^2 + C'$$

$$\leq C''.$$

The final line follows from Assumption (c), that the gradients of the twisting functions are bounded. The above derivation therefore provides that each $w_t$ is bounded, concluding the proof. $\square$

## B  Riemannian Twisted Diffusion Sampler

This section provides additional details on the extension of TDS to Riemannian diffusion models introduced in Section 3. We first introduce the tangent normal distribution. We then provide with background on Riemannian diffusion models, which we parameterize with the tangent normal. Then we describe the extension of TDS to these models. Finally we show how Algorithm 1 modifies to this setting.

**The tangent normal distribution.**  Just as the Gaussian is the parametric family underlying Euclidean diffusion models in Equation (3), the tangent normal (see e.g. [4, 8]) underlies generation in Riemannian diffusion models so we review it here.

We take the tangent normal to be the distribution is implied by a two step procedure. Given a variable $x$ in the manifold, the first step is to sample a variable $\bar{y}$ in $\mathcal{T}_x$, the tangent space at $x$; if $\{h_1, \ldots, h_D\}$ is an orthonormal basis of $\mathcal{T}_x$ one may generate

$$\bar{y} = \mu + \sum_{d=1}^{D} \sigma \epsilon_d \cdot h_d,$$

with $\epsilon_d \overset{i.i.d.}{\sim} \mathcal{N}(0,1)$, and $\sigma^2 > 0$ a variance parameter. The second step is to project $\bar{y}$ back onto the manifold to obtain $y = \exp_x\{\bar{y}\}$ where $\exp_x\{\cdot\}$ denotes the exponential map at $x$. The resulting distribution of $y$ is denoted as $\mathcal{TN}_x(\mu, \sigma^2)$. By construction, $\mathcal{T}_x$ is a Euclidean space with its origin at $x$, so when $\|\mu\|_2 = 0, \mathcal{TN}_x(\mu, \sigma^2)$ is centered on $x$. And since the geometry of a Riemannian manifold is locally Euclidean, when $\sigma^2$ is small the exponential map is close to the identity map and the tangent normal is essentially a narrow Gaussian distribution in the manifold at $x$. Finally, we use $\mathcal{TN}_x(y; \mu, \sigma^2)$ to denote the density of the tangent normal evaluated at $y$.

Because this procedure involves a change of variables from $\bar{y}$ to $y$ (via the exponential map), to compute the tangent normal density one computes

$$\mathcal{TN}_x(y; \mu, \sigma^2) = \mathcal{N}(\exp_x^{-1}\{y\}; \mu, \sigma^2) \left| \frac{\partial}{\partial y} \exp_x^{-1}\{y\} \right|$$

where $\exp_x^{-1}\{y\}$ is the inverse of the exponential map (from the manifold into $\mathcal{T}_x$), and $\left| \frac{\partial}{\partial y} \exp_x^{-1}\{y\} \right|$ is the determinant of the Jacobian of the exponential map; when the manifold lives in a higher dimensional subset of $R$, we take $\left| \frac{\partial}{\partial y} \exp_x^{-1}\{y\} \right|$ to be the product of the positive singular values of the Jacobian.

**Riemannian diffusion models and the tangent normal distribution.** Riemannian diffusion models proceeds through a geodesic random walk [8]. At each step $t$, one first samples a variable $\bar{x}^t$ in $\mathcal{T}_{x^{t+1}}$; if $\{h_1, \ldots, h_D\}$ is an orthonormal basis of $\mathcal{T}_{x^{t+1}}$ one may generate

$$\bar{x}^t = \sigma_{t+1}^2 s_\theta(x^{t+1}) + \sum_{d=1}^{D} \sigma_{t+1} \epsilon_d \cdot h_d,$$

with $\epsilon_d \overset{i.i.d.}{\sim} \mathcal{N}(0,1)$. One then projects $\bar{x}^t$ back onto the manifold to obtain $x^t = \exp_{x^{t+1}}\{\bar{x}^t\}$ where $\exp_x\{\cdot\}$ denotes the exponential map at $x$.

This is equivalent to sampling $x^t$ from a tangent normal distribution as

$$x^t \sim p(x^t \mid x^{t+1}) = \mathcal{TN}_{x^{t+1}}\left(x^t; \sigma_{t+1}^2 s_\theta(x^{t+1}), \sigma_{t+1}^2\right).$$

**TDS for Riemannian diffusion models.** To extend TDS, appropriate analogues of the twisted proposals and weights are all that is needed. For this extension we require that the diffusion model is also associated with a manifold-valued denoising estimate $\hat{x}_\theta$ as will be the case when, for example, $s_\theta(x^t, t) := \nabla_{x^t} \log q(x^t \mid x^0 = \hat{x}_\theta)$ for $\hat{x}_\theta = \hat{x}_\theta(x^t, t)$. In contrast to the Euclidean case, a relationship between a denoising estimate and a computationally tractable score approximation may not always exist for arbitrary Riemannian manifolds; however for Lie groups when the the forward diffusion is the Brownian motion, tractable score approximations do exist [39, Proposition 3.2].

For the case of positive and differentiable $p_{y|x^0}(y|x^0)$, we again choose twisting functions $\tilde{p}_\theta(y \mid x^t) := p_{y|x^0}(y|\hat{x}_\theta(x^t))$.

Next are the inpainting and inpainting with degrees of freedom cases. Here, assume that $x^0$ lives on a multidimensional manifold (e.g. $SE(3)^N$) and the unmasked observation $y = x_{\mathbf{M}}^0$ with $\mathbf{M} \subset \{1, \ldots, N\}$ on a lower-dimensional sub-manifold (e.g. $SE(3)^{|\mathbf{M}|}$, with $|\mathbf{M}| < N$). In this case, twisting functions are constructed exactly as in Section 3.3, except with the normal density in Equation (13) replaced with a Tangent normal as

$$\tilde{p}_\theta(y \mid x^t) = \mathcal{TN}_{\hat{x}_\theta(x^t)_{\mathbf{M}}}(y; 0, \bar{\sigma}_t^2).$$

For all cases, we propose the twisted proposal as

$$\tilde{p}_\theta(x^t \mid x^{t+1}, y) = \mathcal{TN}_{x^{t+1}}\left(x^t; \sigma_{t+1}^2 s_\theta(x^{t+1}, y), \tilde{\sigma}_{t+1}^2\right) \tag{17}$$

where as in the Euclidean case $s_\theta(x^t, y) = s_\theta(x^t) + \nabla_{x^t} \log \tilde{p}(y \mid x^t)$.

Weights at intermediate steps are computed as in the Euclidean case (Equation (11)):

$$w_t(x^t, x^{t+1}) := \frac{p_\theta(x^t \mid x^{t+1})\tilde{p}_\theta(y \mid x^t)}{\tilde{p}_\theta(y \mid x^{t+1})\tilde{p}_\theta(x^t \mid x^{t+1}, y)}$$

$$= \frac{\mathcal{TN}_{x^{t+1}}(x^t; \sigma_{t+1}^2 s_\theta(x^{t+1}), \sigma_{t+1}^2)\tilde{p}_\theta(y \mid x^t)}{\tilde{p}_\theta(y \mid x^{t+1})\mathcal{TN}_{x^{t+1}}(x^t; \sigma_{t+1}^2 s_\theta(x^{t+1}, y), \tilde{\sigma}_{t+1}^2)}.$$

While the proposal and target contribute identical Jacobian determinant terms that cancel out, they remain in the twisting functions.

**Adapting the TDS algorithm to the Riemannian setting.** To translate the TDS algorithm to the Riemannian setting we require only two changes.

The first is on Algorithm 1 Line 6. Here we assume that the unconditional score is computed through the transition density function:

$$s_\theta(x^{t+1}) := \nabla_{x^{t+1}} \log q_{t+1|0}(x^{t+1} \mid \hat{x}_\theta) \quad \text{for } \hat{x}_\theta = \hat{x}_\theta(x^{t+1}).$$

Note that the gradient above ignores the dependence of $\hat{x}_\theta$ on $x^{t+1}$.

The conditional score approximation in Line 6 is then replaced with

$$\tilde{s}_k \leftarrow \tilde{p}_\theta(\cdot \mid x_k^{t+1}, y) := \nabla_{x_k^{t+1}} \log q_{t+1|0}(x_k^{t+1} \mid \hat{x}_\theta) + \nabla_{x_k^{t+1}} \log \tilde{p}_k^{t+1} \quad \text{for } \hat{x}_\theta = \hat{x}_\theta(x_k^{t+1}).$$

Notably, $\tilde{s}_k$ is a $\mathcal{T}_{x_k^{t+1}}$-valued Riemannian gradient.

The second change is to make the proposal on Algorithm 1 Line 7 a tangent normal, as defined in eq. (17).

## C   Additional Related Work

**Training with conditioning information.** Several approaches involve training a neural network to directly sample from a conditional diffusion models. These approaches include (i) conditional training with embeddings of conditioning information, e.g. for denoising images [30], and text-to-image generation [27], (ii) conditioning training with a subset of the state space, e.g. for protein design [37], and image inpainting [29], (iii) classifier-guidance [9, 33], an approach for generating samples of a desired class, which requires training a time-dependent classifier to approximate, for each $t$, $p_\theta(y \mid x^t)$, and training such a time-dependent classifier may be inconvenient and costly), and (iv) classifier-free guidance [17], an approach that builds on classifier-guidance without training a classifier, and instead trains a diffusion model with class information as additional input.

**Langevin and Metropolis-Hastings steps.** Some prior work has explored using Markov chain Monte Carlo steps in sampling schemes to better approximate conditional distributions. For example unadjusted Langevin dynamics [19, 33] or Hamiltonian Monte Carlo [11] in principle permit asymptotically exact sampling in the limit of many steps. However these strategies are only guaranteed to target the conditional distributions of the joint model under the (unrealistic) assumption that the unconditional model exactly matches th forward process, and in practice adding such steps can worsen the resulting samples, presumably as a result of this approximation error [21]. By contrast, TDS does not require the assumption that the learned diffusion model exactly matches the forward noising process.

## D   Empirical results additional details

### D.1   Synthetic diffusion models on two dimensional problems

**Forward process.** Our forward process is variance preserving (as described in Appendix A) with $T = 100$ steps and a quadratic variance schedule. We set $\sigma_t^2 = \sigma_{\min}^2 + (\frac{t}{T})^2 \sigma_{\max}^2$ with $\sigma_{\min}^2 = 10^{-5}$ and $\sigma_{\max}^2 = 10^{-1}$.

**Unconditional target distributions and likelihood.** We evaluate the different methods with two different unconditional target distributions:

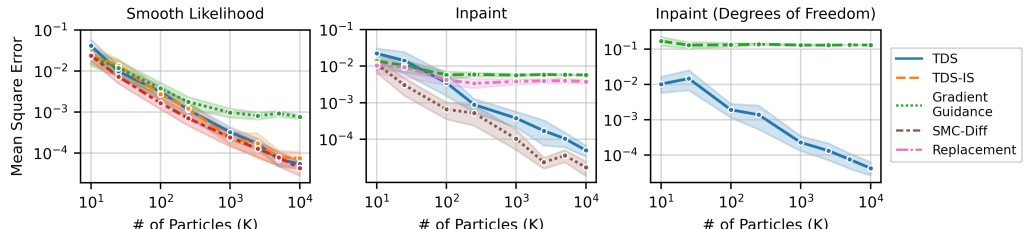

Figure D: Errors of conditional mean estimations with 2 SEM error bars averaged over 25 replicates on mixture of Gaussians unconditional target. TDS applies to all three tasks and provides increasing accuracy with more particles.

1. A **bivariate Gaussian** with mean at $(\frac{1}{2}, \frac{1}{2})$ and covariance 0.9 and
2. A **Gaussian mixture** with three components with mixing proportions $[0.3, 0.5, 0.2]$, means $[(1.54, -0.29), (-2.18, 0.57), (-1.09, -1.40)]$, and 0.2 standard deviations.

We evaluate on conditional distributions defined by the three likelihoods described in Section 5.1.

Figure D provides results analogous to those in Figure 1 but with the mixture of Gaussians unconditional target. In this second example we evaluate a with a variation of the inpainting degrees of freedom case wherein we consider $y = 1$ and $\mathcal{M} = \{[1], [2]\}$, so that $p_{y|x^0}(y|x^0) = \delta_y(x_1^0) + \delta_y(x_2^0)$.

### D.2 Image conditional generation experiments

We perform extensive ablation studies on class-conditional generatoin task and inpainting task using the small-scale MNIST dataset ($28 \times 28 \times 1$ dimensions) in Appendix D.2.1 and Appendix D.2.2. And we show TDS's applicability to higher-dimensional datasets, namely CIFAR10 ($32 \times \times 3$ dimensions) and ImageNet256 ($256 \times 256 \times 3$ dimensions) in Appendix D.2.3.

#### D.2.1 Class-conditional generation on MNIST

**Set-up.** For MNIST, we set up a diffusion model using the variance preserving framework. The model architecture is based on the guided diffusion codebase[5] with the following specifications: number of channels = 64, attention resolutions = "28,14,7", number of residual blocks = 3, learn sigma (i.e. to learn the variance of $p_\theta(x^{t-1} \mid x^t)$) = True, resblock updown = True, dropout = 0.1, variance schedule = "linear". We trained the model for 60k epochs with a batch size of 128 and a learning rate of $10^{-4}$ on 60k MNIST training images. The model uses $T = 1,000$ for training and $T = 100$ for sampling.

The classifier used for class-conditional generation and evaluation is a pretrained ResNet50 model.[6] This classifier is trained on the same set of MNIST training images used by diffusion model training.

In addition we include a variation called *TDS-truncate* that truncates the TDS procedure at $t = 10$ and returns the prediction $\hat{x}_\theta(x^{10})$.

**Sample plots.** To supplement the sample plot conditioned on class 7 in Figure 2a, we present samples conditioned on each of the remaining 9 classes and from other methods. We observe that samples from Gradient Guidance have noticeable artifacts, whereas the other 4 methods produce authentic and correct digits. However, most samples from IS or TDS-IS are identical due to the collapse of importance weights. By contrast, samples from TDS and TDS-truncate have greater diversity, with the latter exhibiting slightly more variations

**Ablation study on twist scales.** We consider exponentiating and re-normalizing twisting functions by a *twist scale* $\gamma$, i.e. setting new twisting functions to $\tilde{p}_\theta(y \mid x^t; \gamma) \propto p_{y|x^0}(y|\hat{x}_\theta(x^t))^\gamma$. In particular, when $t = 0$, we set $p_{y|x^0}(y|x^0; \gamma) \propto p_{y|x^0}(y|x^0)^\gamma$. This modification suggests that the targeted conditional distribution is now

$$p_\theta(x^0 \mid y; \gamma) \propto p_\theta(x^0) p_{y|x^0}(y|x^0)^\gamma.$$

---

[5]https://github.com/openai/guided-diffusion
[6]Downloaded from https://github.com/VSehwag/minimal-diffusion

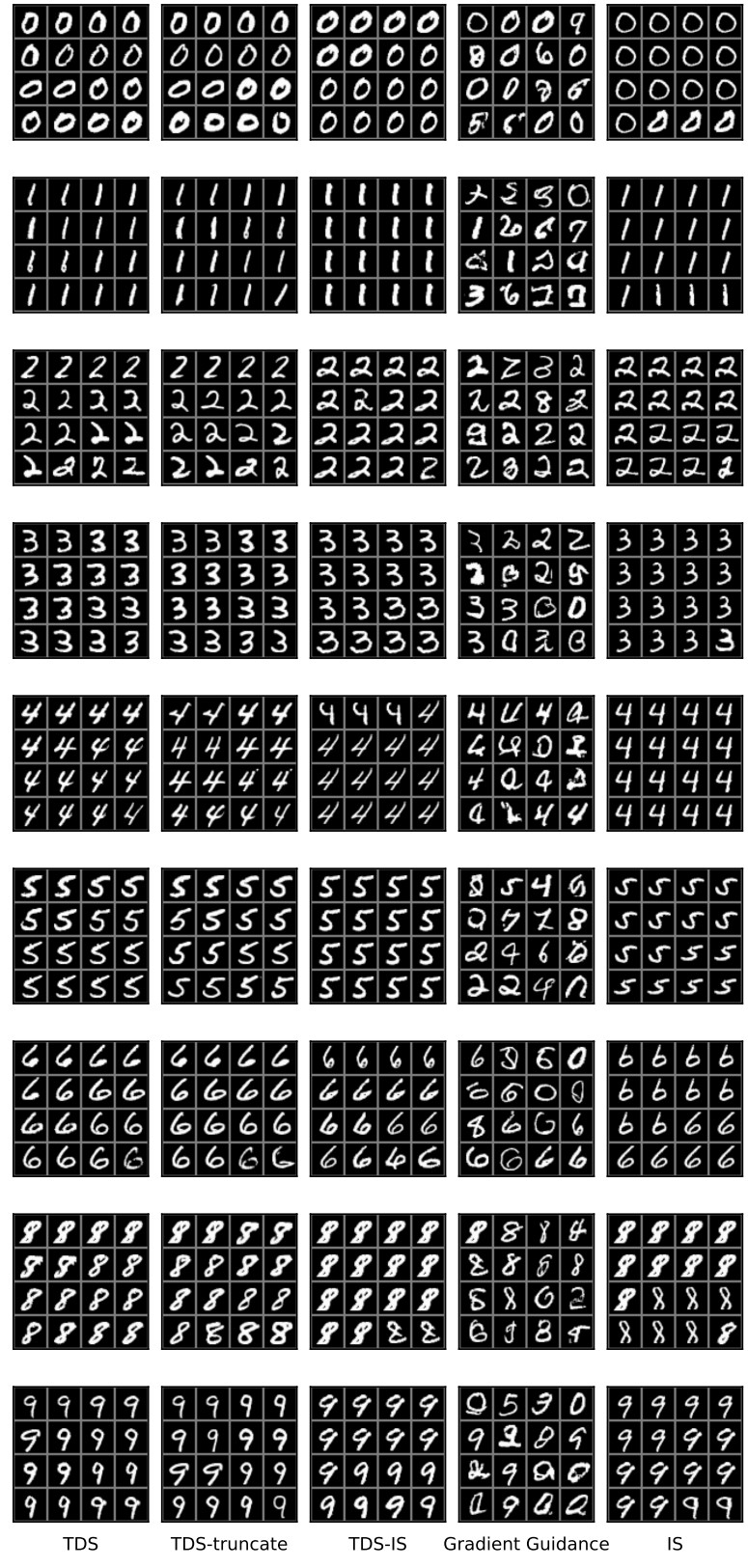

TDS      TDS-truncate      TDS-IS      Gradient Guidance      IS

Figure E: MNIST class-conditional generation. 16 randomly chosen conditional samples from 64 particles in a single run, given class $y$. From top to bottom, $y = 0, 1, 2, 3, 4, 5, 6, 8, 9$.

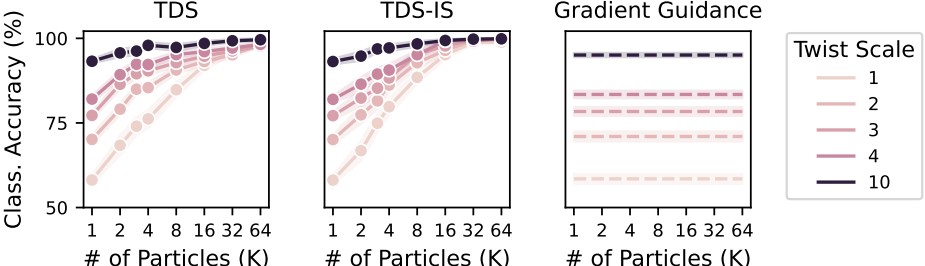

(a) Classification accuracy (measured by the neural network classifier) v.s. number of particles $K$, under different twist scales. Results are averaged over 1,000 random runs with error bands indicating 2 standard errors. Across the panels we see that for all the methods, the larger the twist scale (i.e. the darker the line color), the higher the classification accuracy. For TDS and TDS-IS, this improvement is more significant for smaller value of $K$,

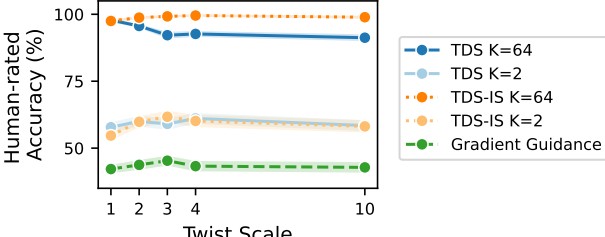

(b) Human-rated classification accuracy v.s. twist scale, for TDS ($K = 64, 2$), TDS-IS ($K = 64, 2$) and Gradient Guidance. Results are averaged over 640 randomly chosen samples with error bands indicating 2 standard errors. For TDS ($K = 64$), increasing the twist scale generally decreases the human-rated accuracy. For remaining methods, a moderate increase in twist scale improves the human-rated accuracy; however, excessively large twist scale can hurt the accuracy.

Figure F: MNIST class-conditional generation: classification accuracy under different twist scales, computed by a neural network classifier (top panel) and a human (bottom panel).

By setting $\gamma > 1$, the classification likelihood becomes sharper, which is potentially helpful for twisting the samples towards a specific class. The TDS algorithm (and likewise TDS-IS and Gradient Guidance) still apply with this new definition of twisting functions. The use of twist scale is similar to the *guidance scale* introduced in the literature of Classifier Guidance, which is used to multiply the gradient of the log classification probability [9].

In Figure F, we examine the effect of varying twist scales on classification accuracy of TDS, TDS-IS and Gradient Guidance. We consider two ways to evaluate the accuracy. First, *classification accuracy* computed by a *neural network classifier*, where the evaluation setup is the same as in Section 5.2. Second, the *human-rated* classification accuracy, where a human (one of the authors) checks if a generated digit has the right class and does not have artifacts. Since human evaluation is expensive, we only evaluate TDS, TDS-IS (both with $K = 64, 2$) and Gradient Guidance. In each run, we randomly sample one particle out of $K$ particles according to the associated weights. We conduct 64 runs for each class label, leading to a total of 640 samples for human evaluation.

Figure Fa depicts the classification accuracy measured by a neural network classifier. We observe that in general larger twist scale improves the classification accuracy. For TDS and TDS-IS, the improvement is more significant for smaller number of particles $K$ used.

Figure Fb depicts the human-rated accuracy. In this case, we find that larger twist scale is not necessarily better. A moderately large twist scale ($\gamma = 2, 3$) generally helps increasing the accuracy, while an excessively large twist scale ($\gamma = 10$) decreases the accuracy. An exception is TDS with $K = 64$ particles, where any $\gamma > 1$ leads to worse accuracy compared to the case of $\gamma = 1$. Study on twist scale aside, we find that using more particles $K$ help improving human-rated accuracy (recall that Gradient Guidance is a special case of TDS with $K = 1$): given the same twist scale, TDS with $K = 1, 2$ or 64 has increasing accuracy. In addition, both TDS and TDS-IS with $K = 64$ and $\gamma = 1$

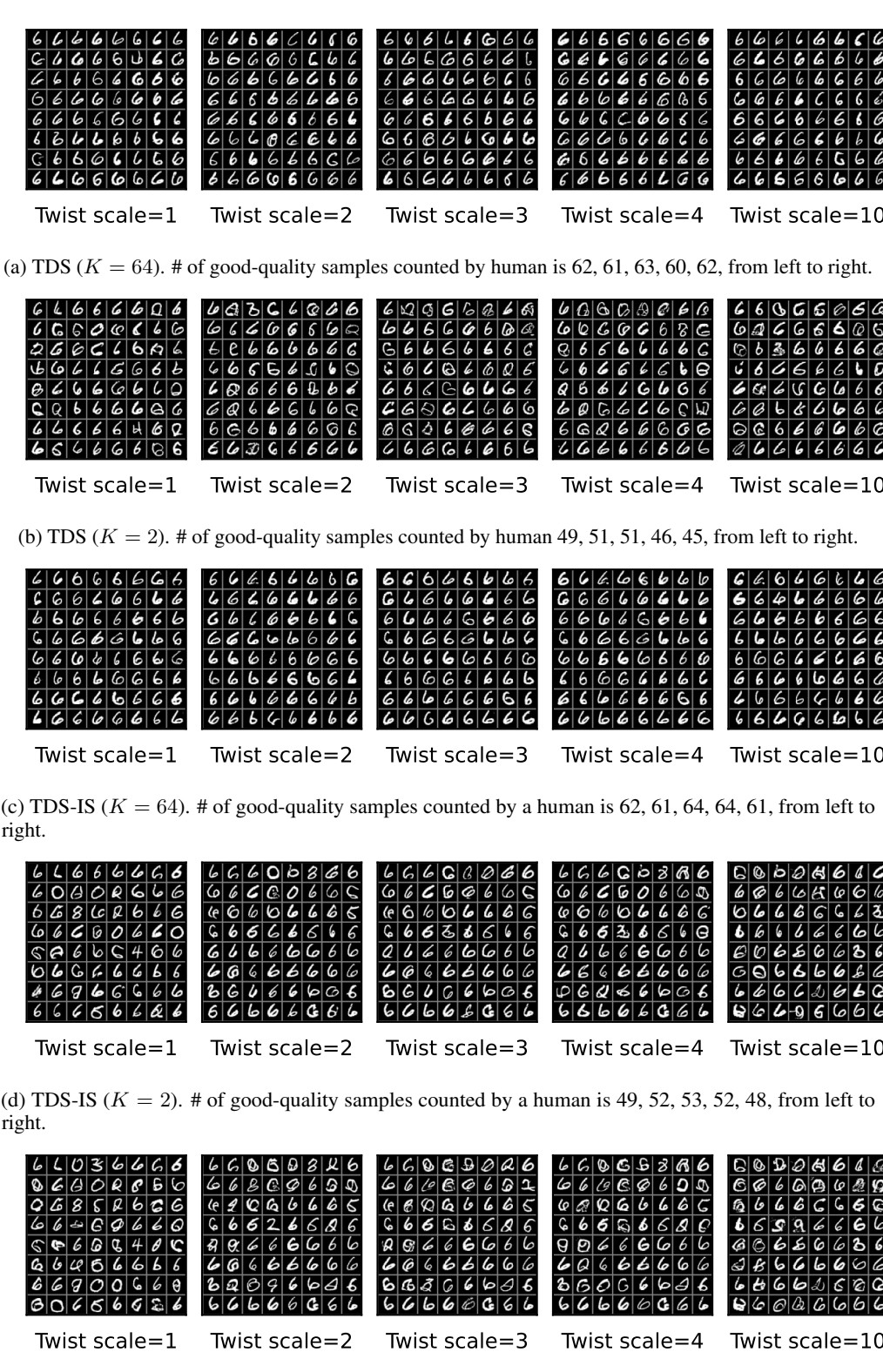

(a) TDS ($K = 64$). # of good-quality samples counted by human is 62, 61, 63, 60, 62, from left to right.

(b) TDS ($K = 2$). # of good-quality samples counted by human 49, 51, 51, 46, 45, from left to right.

(c) TDS-IS ($K = 64$). # of good-quality samples counted by a human is 62, 61, 64, 64, 61, from left to right.

(d) TDS-IS ($K = 2$). # of good-quality samples counted by a human is 49, 52, 53, 52, 48, from left to right.

(e) Gradient Guidance. # of good-quality samples counted by a human is 31, 38, 41, 39, 34, from left to right.

Figure G: MNIST class-conditional generation: random samples selected from 64 random runs conditioned on class 6 under different twist scales. Top to bottom: TDS ($K = 64$), TDS ($K = 2$), TDS-IS ($K = 64$), TDS-IS ($K = 2$) and Gradient Guidance. In general, moderately large twist scales improve the sample quality. However, overly large twist scale (e.g. 10) would distort the digit shape with more artifacts, though retaining useful features that may allow a neural network classifier to identity the class.

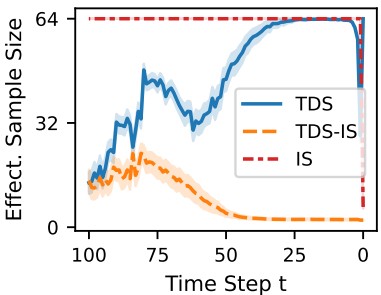

Figure H: ESS trace avg. over 100 runs ($K = 64$)

have almost perfect accuracy. The effect of $K$ on human-rated accuracy is consistent with previous findings with neural network classifier evaluation in Section 5.2.

We note that there is a discrepancy on the effects of twist scales between neural network evaluation and human evaluation. We suspect that when overly large twist scale is used, the generated samples may fall out of the data manifold; however, they may still retain features recognizable to a neural network classifier, thereby leading to a low human-rated accuracy but a high classifier-rated accuracy. To validate this hypothesis, we present samples conditioned on class 6 in Figure G. For example, in Figure Ge, Gradient Guidance with $\gamma = 1$ has 31 good-quality samples out of 64, and the rest of the samples often resamble the shape of other digits, e.g. 3,4,8; and Gradient Guidance with $\gamma = 10$ has 34 good-quality samples, but most of the remaining samples resemble 6 with many artifacts.

**Effective sample size.** Effective sample size (ESS) is a common metric used to diagnose the performance of SMC samplers, which is defined as $(\sum_{k=1}^{K} w_k^t)^2 / (\sum_{k=1}^{K} (w_k^t)^2)$ for $K$ weighted particles $\{x_k^t, w_k^t\}_{k=1}^{K}$. Note that ESS is always bounded between 0 and $K$.

Figure H depicts the ESS trace comparison of TDS, TDS-IS, and IS. TDS has a general upward trend of ESS approaching $K = 64$. Though in the final few steps ESS of TDS drops by a half, it is still higher than that of TDS-IS and IS that deteriorates to around 1 and 6 respectively. In practice, one can use the TDS-truncate variant introduced above to avoid the ESS drop in the final few steps.

### D.2.2 Image inpainting on MNIST

The inpainting task is to sample images from $p_\theta(x^0 \mid x_{\mathbf{M}}^0 = y)$ given observed part $x_{\mathbf{M}}^0 = y$. Here we segment $x^0 = [x_{\mathbf{M}}^0, x_{\overline{\mathbf{M}}}^0]$ into observed dimensions $\mathbf{M}$ and unobserved dimensions $\overline{\mathbf{M}}$, as described in Section 3.3.

In this experiment, we consider two types of observed dimensions: (1) $\mathbf{M}$ = "half", where the left half of an image is observed, and (2) $\mathbf{M}$ = "quarter", where the upper left quarter of an image is observed.

We run TDS, TDS-IS, Gradient Guidance, SMC-Diff, and Replacement method to inpaint 10,000 validation images. We also include TDS-truncate that truncates the TDS procedure at $t = 10$ and returns $\hat{x}_\theta(x^{10})$.

**Twisting function variance schedule.** We use a flexible variance schedule $\{\hat{\sigma}_t^2\}$ to extend the twisting functions in eq. (13) to

$$\tilde{p}_\theta(y \mid x^t, \mathbf{M}) \coloneqq \mathcal{N}(y \mid \hat{x}_\theta(x^t)_{\mathbf{M}}, \hat{\sigma}_t^2).$$

Ideally, $\hat{\sigma}_t^2$ should match $\mathrm{Var}_{p_\theta}[y \mid x^t, \mathbf{M}]$. In TDS, we choose $\hat{\sigma}_t^2 = \frac{(\bar{\sigma}_t^2/\bar{\alpha}_t) \cdot \tau^2}{\bar{\sigma}_t^2/\bar{\alpha}_t + \tau^2}$ where $\tau^2 = 0.12$ is the estimated sample variance of the training data (averaged over pixels). This choice is inspired from Song et al. [32, Appendix A.3]: if the data distribution is $q(x^0) = \mathcal{N}(0, \tau^2)$, and $q(x^t \mid x^0) = \mathcal{N}(x^t \mid \sqrt{\bar{\alpha}_t} x^0, \sigma_t^2)$, then by Bayes rule $\mathrm{Var}_q[x^0 \mid x^t] = \frac{(\bar{\sigma}_t^2/\bar{\alpha}_t) \cdot \tau^2}{(\bar{\sigma}_t^2/\bar{\alpha}_t) + \tau^2}$.

Changing $\hat{\sigma}_t^2$ has a similar effect as using a twist scale to exponentiate the twisting functions in Appendix D.2.1. In particular, for a Gaussian distribution, exponentiating the distribution corresponds to re-scaling the variance: $\mathcal{N}(x \mid \mu, (\sigma/\sqrt{\gamma})^2) \propto \mathcal{N}(x \mid \mu, \sigma^2)^\gamma$.

**Metrics.** We consider 3 metrics. (1) We use the effective sample size (ESS) to compare the particle efficiency among different SMC samplers (namely TDS, TDS-IS, and SMC-Diff).

(2) In addition, we ground the performance of a sampler in the downstream task of classifying a partially observed image $x_{\mathbf{M}}^0 = y$: we use a classifier to predict the class $\hat{z}$ of an inpainted image $x^0$, and compute the accuracy against the true class $z^*$ of the unmasked image ($z^*$ is provided in MNIST dataset). This prediction is made by the same classifier used in Section 5.2.

Consider $K$ weighted particles $\{x_k^0; w_k^0\}_{k=1}^K$ drawn from a sampler conditioned on $x_{\mathbf{M}}^0 = y$ and assume the weights are normalized. We define the *Bayes accuracy* (BA) as

$$\mathbb{1}\left\{\hat{z}(y) = z^*\right\}, \quad \text{with} \quad \hat{z}(y) := \underset{z=1:10}{\arg\max} \sum_{k=1}^K w_k^0 p(z; x_k^0),$$

where $\hat{z}(y)$ is viewed as an approximation to the Bayes optimal classifier $\hat{z}^*(y)$ given by

$$\hat{z}^*(y) := \underset{z=1:10}{\arg\max}\, p_\theta(z \mid x_{\mathbf{M}}^0 = y) = \underset{z=1:10}{\arg\max} \int p_\theta(x^0 \mid x_{\mathbf{M}}^0 = y)p(z; x^0)dx^0. \qquad (18)$$

(In eq. (18) we assume the classifier $p_{y|x^0}(\cdot|x^0)$ is the optimal classifier on full images.)

(3) We also consider *classification accuracy* (CA) defined as the following

$$\sum_{k=1}^K w_k^0 \mathbb{1}\{\hat{z}(x_k^0) = z^*\}, \quad \text{with} \quad \hat{z}(x_k^0) := \underset{z=1:10}{\arg\max}\, p_{y|x^0}(z|x_k^0).$$

BA and CA evaluate different aspects of a sampler. BA is focused on the optimal prediction among multiple particles, whereas CA is focused on the their weighted average prediction.

**Comparison results of different methods.** Figure I depicts the ESS trace, BA and CA for different samplers. The overall observations are similar to the observations in the class-conditional generation task in Section 5.2, except that SMC-Diff and Replacement methods are not available there.

Replacement has the lowest CA and BA across all settings. Comparing TDS to SMC-Diff, we find that SMC-Diff's ESS is consistently greater than TDS; however, SMC-Diff is outperformed by TDS in terms of both CA and BA.

We also note that despite Gradient Guidance's CA is lower, its BA is comparable to TDS. This result is due to that as long as Gradient Guidance generates a few good-quality samples out of $K$ particles, the optimal prediction can be accurate, thereby resulting in a high BA.

**Ablation study on twisting function variance schedule.** We compare the following three options:

1. DPS: $\hat{\sigma}_t := 2\|\hat{x}_\theta(x^t)_{\mathbf{M}} - y\|_2 \sigma_t^2$, adapted from the DPS method [6, Algorithm 1],
2. $\Pi$GDM: $\hat{\sigma}_t := \sigma_t^2/\sqrt{\bar{\alpha}_t}$, adapted from the $\Pi$GDM method [32, Algorithm 1],
3. TDS: $\hat{\sigma}_t^2 = \frac{(\bar{\sigma}_t^2/\bar{\alpha}_t)\cdot\tau^2}{(\bar{\sigma}_t^2/\bar{\alpha}_t)+\tau^2}$ where $\tau^2 = 0.12$.

Figure J shows the classification accuracy of TDS, TDS-IS and Gradient Guidance with different twist scale schemes. We find that our choice has similar performance to that of $\Pi$GDM, and outperforms DPS in most cases. Exceptions are when $\mathbf{M}$ = "quarter" and for large $K$, TDS with twist scale choice of $\Pi$GDM or DPS has higher CA, as is shown in the left panel in Figure Jb.

### D.2.3  Class-conditional generation on ImageNet and CIFAR-10

**ImageNet.** We compare TDS (with # of particles $K = 1, 16$) to Classifier Guidance (CG) using the same unconditional model from `https://github.com/openai/guided-diffusion/tree/main`. CG uses a classifier trained on noisy inputs taken from this repository as well. For TDS, we

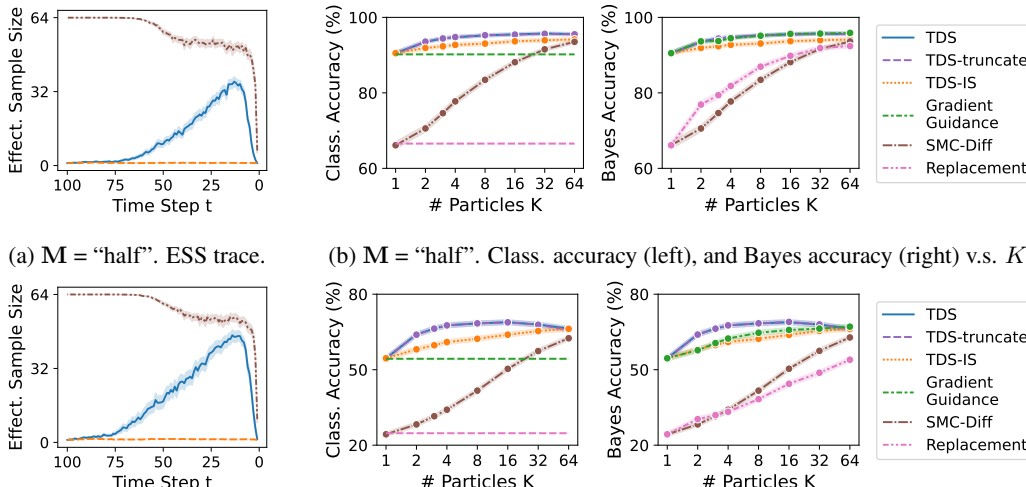

(a) **M** = "half". ESS trace.

(b) **M** = "half". Class. accuracy (left), and Bayes accuracy (right) v.s. $K$.

(c) **M** = "quarter". ESS trace.

(d) **M** = "quarter". Class. accuracy (left), and Bayes accuracy (right) v.s. $K$.

Figure I: MNIST image inpainting. Results for observed dimension **M** = "half" are shown in the top panel, and **M** = "quarter" in the bottom panel. (i) Left column: ESS traces are averaged over 100 different images inpainted by TDS, TDS-IS, and SMC-Diff, all with $K = 64$ particles. TDS's ESS is generally increasing untill the final 20 steps where it drops to around 1, suggesting significant particle collapse in the end of generation trajectory. TDS-IS's ESS is always around 1. SMC-Diff has higher particle efficiency compared to TDS. (ii) Right column: Classification accuracy and Bayes accuracy are averaged over 10,000 images. In general increasing the number of particles $K$ would improve the performance of all samplers. TDS and TDS-truncate have the highest accuracy among all given the same $K$. (iii) Finally, comparison of top and bottom panels shows that in a harder inpainting problem where **M** = "quarter", TDS's has a higher ESS but lower CA and BA.

| Method | Classification Accuracy ↑ | FID ↓ | Inception Score ↑ |
|---|---|---|---|
| TDS ($K = 16$) | 99.90% | 26.65 | 64.03 |
| TDS ($K = 1$) | 99.63% | 26.05 | 44.45 |
| Classifier Guidance | 99.17% | 14.03 | 100.33 |
| Unconditional model | n/a | 26.21 | 39.70 |

Table 1: ImageNet. Comparison of sample quality. The top three methods are evaluated on 16k samples generated with 100 steps. The unconditional model performance is provided in [9], which is evaluated on 50k samples generated with 250 steps.

use the same classifier evaluated at timestep = 0 to mimic a standard trained classifier. We generate 16 images for each of the 1000 class labels, using 100 sampling steps. TDS uses a twist scale of 10, and CG uses a guidance scale of 10. Notably, given a fixed class, TDS ($K = 16$) generates correlated samples in a single SMC run, and TDS ($K = 1$) and CG generate 16 independent samples.

In Figure K, we observe that TDS can faithfully capture the class and have comparable image quality to CG's , although with less diversity than CG and TDS ($K = 1$).

Figure L shows more samples given randomly selected classes. We also reported results of the unconditional model from the original paper [9] that are evaluated on 50k samples with 250 sampling steps in Appendix D.2.3. TDS and CG provide similar classification accuracy. TDS has similar FIDs compared to the unconditional model and better inception score. CG's FID and inception score are better than TDS. We suspect this difference is attributed to the sample correlation (and hence less diversity) within particles in a single run of TDS ($K = 16$).

**CIFAR-10.** We ran TDS ($K = 16$) with the twist scale = 1 and 100 sampling steps using diffusion model from `https://github.com/openai/improved-diffusion` and classifier from `https://github.com/VSehwag/minimal-diffusion/tree/main`. TDS generates faithful and diverse

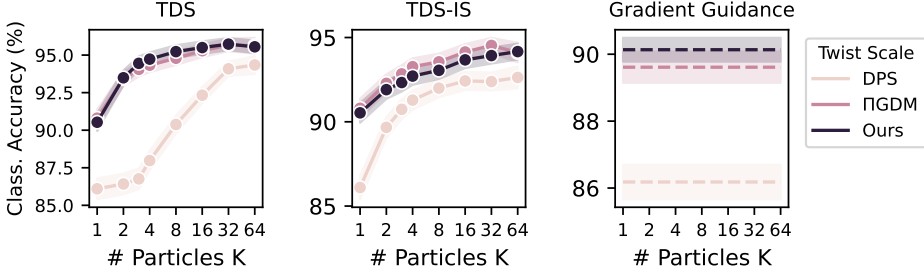

(a) **M** = "half". Classification accuracy under different twist scale schemes.

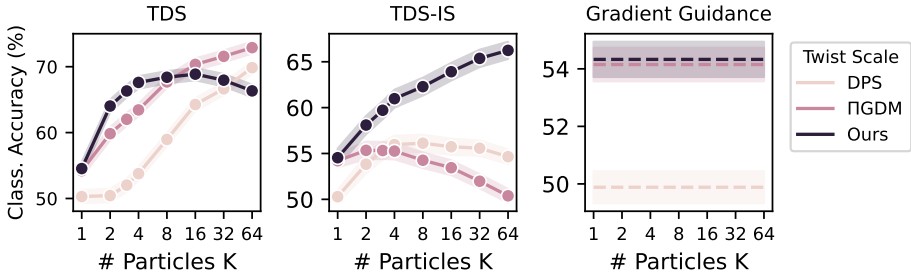

(b) **M** = "quarter". Classification accuracy under different twist scale schemes.

Figure J: MNIST image inpainting: Ablation study on twist scales. Top: observed dimensions **M** = "half". Bottom: **M** = "quarter". In most case, the performance of our choice of twist scale is similar to that of $\Pi$GDM, and is better compared to DPS.

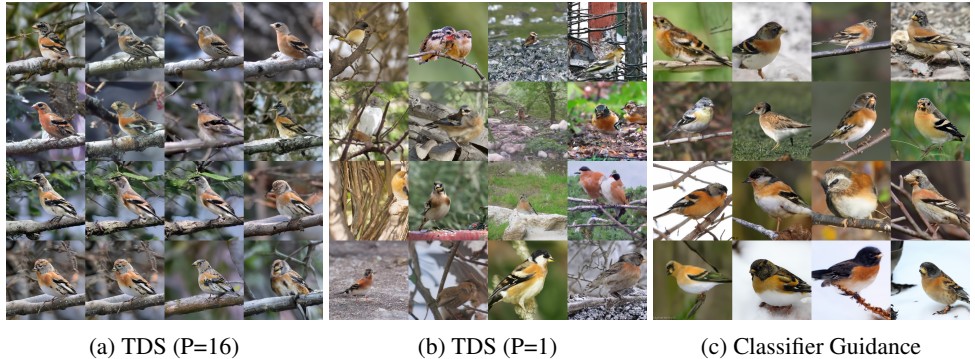

(a) TDS (P=16)       (b) TDS (P=1)       (c) Classifier Guidance

Figure K: 256×256 ImageNet. Samples from TDS and Classifier Guidance given class 'brambling'.

images, see Figure M. However, we found TDS can occasionally generate incorrect samples for the class 'truck' (the last panel in Figure M).

Finally, we present the ESS trace plot of the three image datasets in Figure N. We see there is a sudden ESS drop in the final stage for MINST, sometimes for CIFAR10, but usually not on ImageNet. Mechanically, the drop in ESS implies a large discrepancy between the final and intermediate target distributions. We suspect such discrepancies might arise from irregularities in the denoising network near $t = 0$. In practice, one can consider early truncating the TDS procedure to prevent such ESS drops and promote particle diversity, as is studied in Appendices D.2.1 and D.2.2.

# E   Motif-scaffolding application details

**Unconditional model of protein backbones.** We here use the FrameDiff, a diffusion generative model described by Yim et al. [39]. FrameDiff parameterizes protein $N$-residue protein backbones as a collection of rigid bodies defined by rotations and translations as $x^0 = [(R_1, z_1), \ldots, (R_N, z_N)] \in$

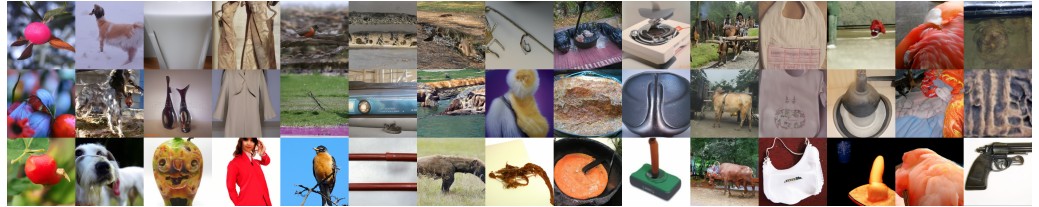

Figure L: 256×256 ImageNet. 15 random class samples. Top to bottom: TDS(P=16), TDS(P=1), Classifier Guidance.

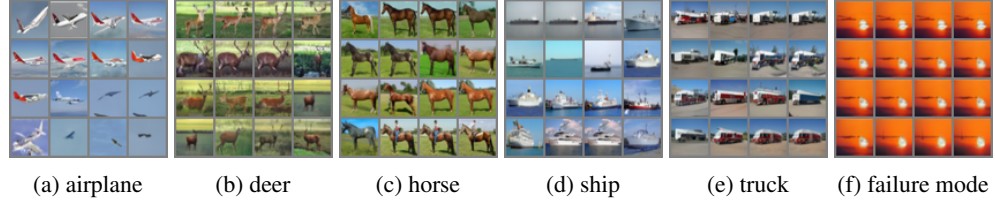

    (a) airplane        (b) deer        (c) horse        (d) ship        (e) truck    (f) failure mode

Figure M: CIFAR10. Samples from TDS with $K = 16$ particles given different class labels (within a single SMC run). In most cases TDS generate authentic and diverse images. The last panel presents a failure mode for the class 'truck'.

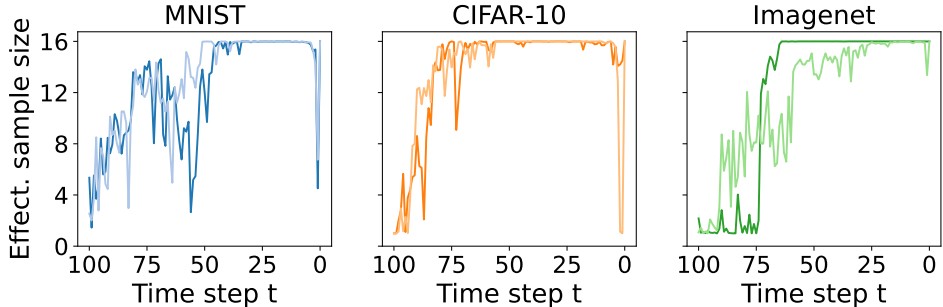

Figure N: 2 example effective sample size traces of TDS ($K = 16$, 100 sampling steps) on MNIST, CIFAR-10 and ImageNet models, respectively.

$SE(3)^N$. $SE(3)$ is the special Euclidean group in three dimensions (a Riemannian manifold). Each $R_n \in SO(3)$ (the special orthogonal group in three dimensions) is a rotation matrix and each $z_n \in \mathbb{R}^3$ in a translation. Together, $R_n$ and $z_n$ describe how one obtains the coordinates of the three backbone atoms C, $C_\alpha$, and N for each residue by translating and rotating the coordinates of an *idealized* residue with $C_\alpha$ carbon and the origin. The conditioning information is then a motif $y = x_{\mathbf{M}}^0 \in SE(3)^{|\mathbf{M}|}$ for some $\mathbf{M} \subset \{1, \ldots, N\}$. FrameDiff is a continuous time diffusion model and includes the number of steps as a hyperparmeter; we use 200 steps in all experiments. We refer the reader to [39] for details on the neural network architecture, and details of the forward and reverse diffusion processes.

**Twisting functions for motif scaffolding.** To define a twisting function, we use a tangent normal approximation to $p_\theta(y \mid x^t)$ as introduced in eq. (15). In this case the tangent normal factorizes across each residue and across the rotational and translational components. The translational components are represented in $\mathbb{R}^3$, and so are treated as in the previous Euclidean cases, using the variance preserving extension described in Appendix A. For the rotational components, represented in the special orthogonal group $SO(3)$, the tangent normal may be computed as described in Appendix B, using the known exponential map on $SO(3)$. In particular, we use as the twisting function

$$\tilde{p}_\theta(y \mid x^t, \mathbf{M}) = \prod_{m=1}^{|\mathbf{M}|} \mathcal{N}(y_m^T; \hat{x}_\theta(x^t)_{\mathbf{M}_m}^T, 1 - \bar{\alpha}_t) \mathcal{TN}_{\hat{x}_\theta(x^t)_{\mathbf{M}_m}^R}(y_m^R; 0, \bar{\sigma}_t^2),$$

where $y_m^T, \hat{x}_\theta(x^t)_{\mathbf{M}_m}^T \in \mathbb{R}^3$ represent the translations associated with the $m^{th}$ residue of the motif and its prediction from $x^t$, and $y_m^R, \hat{x}_\theta(x^t)_{\mathbf{M}_m}^R \in SO(3)$ represent the analogous quantities for the rotational component. Next, $1 - \bar{\alpha}_t = \text{Var}_q[x^t \mid x^0]$ and $\bar{\sigma}_t^2$ is the time of the Brownian motion associated with the forward process at step $t$ [39]. For further simplicity, our implementation further approximates log density of the tangent normal on rotations using the squared Frobenius norm of the difference between the rotations associated with the motif and the denoising prediction (as in [37]), which becomes exact in the limit that $\bar{\sigma}_t$ approaches 0 but avoids computation of the inverse exponential map and its Jacobian.

**Motif rotation and translation degrees of freedom.** We similarly seek to eliminate the rotation and translation of the motif as a degree of freedom. We again represent our ambivalence about the rotation of the motif with randomness, augment our joint model to include the rotation with a uniform prior, and write

$$p_{y|x^0}(y|x^0) = \int p(R) p_{y|x^0}(y|x^0, R) \quad \text{for} \quad p_{y|x^0}(y|x^0, R) = \delta_{Ry}(x^0)$$

and with $p(R)$ the uniform density (Haar measure) on SO(3), and $Ry$ represents rotating the rigid bodies described by $y$ by $R$. For computational tractability, we approximate this integral with a Monte Carlo approximation defined by subsampling a finite number of rotations, $\mathcal{R} = \{R_k\}$ (with $|\mathcal{R}| =$# `Motif Rots.` in total). Altogether we have

$$\tilde{p}_\theta(y \mid x^t) = |\mathcal{R}|^{-1} \cdot |\mathcal{M}|^{-1} \sum_{R \in \mathcal{R}} \sum_{\mathbf{M} \in \mathcal{M}} \tilde{p}_\theta(Ry \mid x^t, \mathbf{M}).$$

The subsampling above introduces the number of motif locations and rotations subsampled as a hyperparameter. Notably the conditional training approach does not immediately address these degrees of freedom, and so prior work randomly sampled a single value for these variables [34, 36, 37].

**Evaluation details.** We use *self-consistency* evaluation approach for generated backbones [34] that (i) uses fixed backbone sequence design (inverse folding) to generate a putative amino acid sequence to encode the backbone, (ii) forward folds sequences to obtain backbone structure predictions, and (iii) judges the quality of initial backbones by their agreement (or self-consistency) with predicted structures. We inherit the specifics of our evaluation and success criteria set-up following [37], including using ProteinMPNN [7] for step (i) and AlphaFold [20] on a single sequence (no multiple sequence alignment) for (ii).

In this evaluation we use ProteinMPNN [7] with default settings to generate 8 sequences for each sampled backbone. Positions not indicated as resdesignable in [37, Methods Table 9] are held fixed. We use AlphaFold [20] for forward folding. We define a "success" as a generated backbone for which at least one of the 8 sequences has backbone atom with both scRMSD < 1 Å on the motif and scRMSD < 2 Å on the full backbone.

We benchmarked TDS on 24/25 problems in the benchmark set introduced [37, Methods Table 9]. A 25th problem (6VW1) is excluded because it involves multiple chains, which cannot be represented by FrameDiff. FrameDiff requires specifying a total length of scaffolds. In all replicates, we fixed the scaffold the median of the `Total Length` range specified by Watson et al. [37, Methods Table 9]. For example, 116 becomes 125 and 62-83 becomes 75.

As discussed in the main text, in our handling of the motif-placement degree of freedom we restrict the number of possible placements considered for discontiguous motifs. To select these placements we (i) restrict to masks which place the motif indices in a pre-specified order that we do not permute and do not separate residues that appear contiguously in source PDB file, and (ii) when there are still too many possible placements sub-sample randomly to obtain the set of masks $\mathcal{M}$ of at most some maximum size (# `Motif Locs.`). We do not enforce the spacing between segments specified in the "contig" description described by [37].

**Resampling.** In all motif-scaffolding experiments, we use systematic resampling. An effective sample size threshold is used $K/2$ in all cases, that is, we trigger resampling steps only when the effective sample size falls below this level.

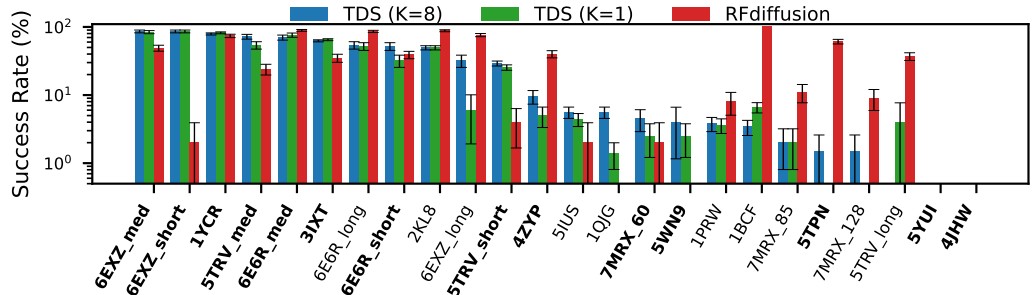

Figure O: Results on full motif-scaffolding benchmark set. The y-axis is the fraction of successes across at least 200 replicates. Error bars are $\pm 2$ standard errors of the mean. Problems with scaffolds shorter than 100 residues are bolded. TDS (K=8) outperforms the state-of-the-art (RFdiffusion) on most problems with short scaffolds.

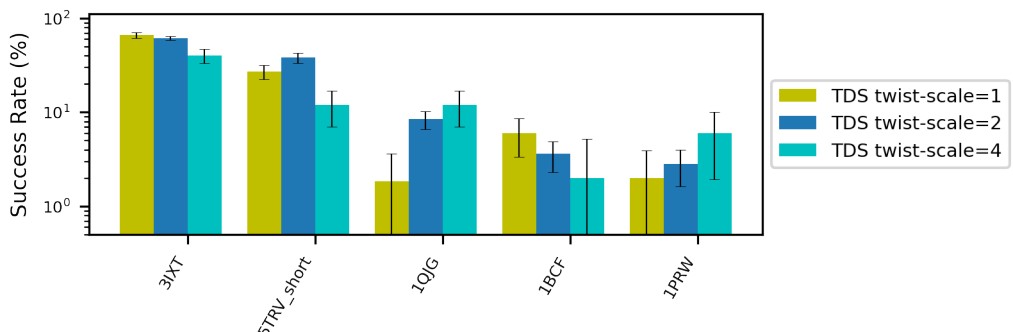

Figure P: Increasing the twist scale has a different impact across motif-scaffolding benchmark problems.

## E.1 Additional Results

**Impact of twist scale on additional motifs.** Figure 3a showed monotonically increasing success rates with the twist scale. However, this trend does not hold for every problem. Figure P demonstrates this by comparing the success rates with different twist-scales on five additional benchmark problems.

**Effective sample size varies by problem.** Figure Q shows two example effective sample size traces over the course of sample generation. For 6EXZ-med resampling was triggered 38 times (with 14 in the final 25 steps), and for 5UIS resampling was triggered 63 times (with 13 in the final 25 steps). The traces are representative of the larger benchmark.

**Application of TDS to RFdiffusion:** We also tried applying TDS to RFdiffusion [37]. RFdiffusion is a diffusion model that uses the same backbone structure representation as FrameDiff. However, unlike FrameDiff, RFdiffusion trained with a mixture of conditional examples, in which a segment of the backbone is presented as input as a desired motif, and unconditional training examples in which the only a noised backbone is provided. Though in our benchmark evaluation provided the motif information as explicit conditioning inputs, we reasoned that TDS should apply to RFdiffusion as well (with conditioning information not provided as input). However, we were unable to compute numerically stable gradients (with respect to either the rotational or translational components of the backbone representation); this lead to twisted proposal distributions that were similarly unstable and trajectories that frequently diverged even with one particle. We suspect this instability owes to RFdiffusion's structure prediction pretraining and limited fine-tuning, which may allow it to achieve good performance without having fit a smooth score-approximation.

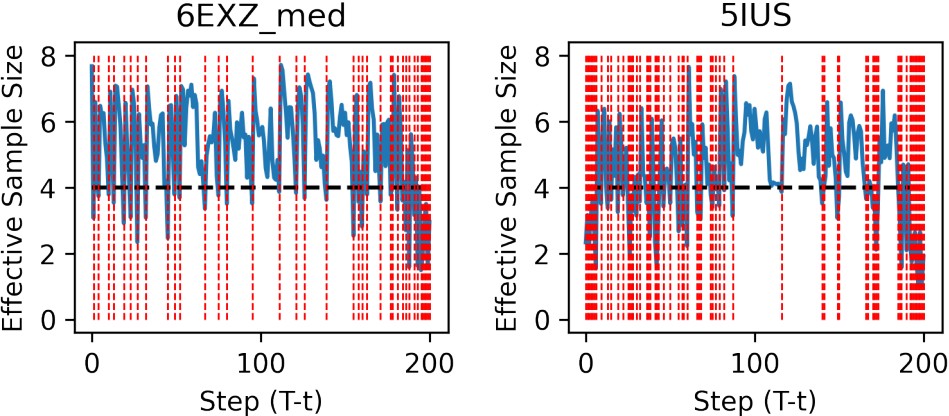

Figure Q: Effective sample size traces for two motif-scaffolding examples (Left) `6EXZ-med` and (Right) `5IUS`. In both case $K = 8$ and a resampling threshold of $0.5K$ is used. Dashed red lines indicate resampling times.

