# OpenReview forum: "Practical and Asymptotically Exact Conditional Sampling in Diffusion Models"
_NeurIPS.cc/2023/Conference — NeurIPS 2023 poster_

### Official Review · Reviewer_BFYS · 2023-07-06

**Soundness:** 3 good
**Presentation:** 4 excellent
**Contribution:** 3 good
**Rating:** 6
**Confidence:** 4

**Summary:**

This paper focuses on solving inverse problems using diffusion based probabilistic models. More precisely, it is only assumed that one has access to a diffusion model for the prior distribution and a likelihood, so that no additional training is needed. The aim of the present paper is to provide an asymptotically exact method.

The method builds up on the idea of reconstruction guidance which aims at approximating the score involved in the diffused posterior,  $\nabla \log p_t(x_t | y)$ which is equal to $\nabla \log p(y|x_t) + \nabla \log p_t(x_t)$ where the second term is given by the prior diffusion model. The first term is however intractable but can be approximated by noting that $p(y | x_t) = \int p(y|x_0) p(x_0 | x_t) dx_0$. This integral is then approximated by simply taking the mean a posteriori i.e. $p(y | x_t) \approx p(y | E(x_0 | x_t))$ which  itself can be approximated using Tweedie's formula and the prior score.

The authors use this idea to obtain particle approximations of the marginal of the backward diffusion of the posterior. They do so using twisting functions. This allows them to define a principled SMC samplers for the target distribution. By standard SMC results, the obtained particle approximation of the posterior converges to the posterior of the diffusion model. The extension to Riemannian manifolds is also provided.


**Strengths:**

- The contribution of this paper is original. Unlike other methods trying to solve inverse problems, this method is theoretically grounded and is guaranteed to be reliable in relatively complex problems. This is for example not the case of DPS [1] which fails to sample the posterior even in this simplest Bayesian settings.

- I also acknowledge that the paper is well written, I enjoyed reading it.

- The numerical experiments are sound and the application to protein design is interesting.

**Weaknesses:**

- While I find the approximations of the optimal twisting functions reasonable, I am not sure if this is the best idea if we take into account the computational cost. Indeed, sampling from the transition requires computing the gradient of the score network with respect to the input. In large scale applications (like images with 3x256x256 dimensions) this severly limits the number of particles that can be used, since the gradient will be computed for each particle. At the end of the day, the number of particles used is very important even if the transition kernels and weights are near optimal; if the posterior is highly multimodal then we inevitably need many particles in order to populate each mode. SMCDiff on the other hand should work better on these problems since one can use a larger number of particles given a fixed computational budget.

- The comparison with SMCDiff is in my opinion unfair. SMCDiff performs particle filtering only on the unobserved part of the state and does not require computing the gradient of the score network. Therefore the computational time and memory cost of TDS are much larger than SMCDiff. This is not taken into account in the numerical experiments. Furthermore, it seems that SMCDiff outperforms TDS in terms of ESS. I understand that SMCDiff relies on a stringent assumption that does not hold in practice, but in my opinion it is as reasonable as saying that the instrumental kernel and weights proposed for TDS are a good approximation. In short, i am not really convinced that TDS is better than SMCDiff for inpainting tasks. For noisy inverse problems of course the comparison is not relevant since SMCDiff is not designed for such problems.

- As far as I can tell, the impact of the dimension of the observation is not discussed. As is widely known in the SMC community, this makes the weights degenerate. I believe that a discussion on this matter should be added so that readers are not misled into thinking that in high dimensional problems this method will provide $K$ diversified samples from the posterior. It should be emphasized that this method works best on low dimensional problems.




**Questions:**

It is claimed in the appendix that the assumptions required to prove the convergence are mild. However, assuming that the ratio of the twisting function (assumption (b), line 588-589) is bounded is not a mild assumption since it doesn't hold in the simplest cases. It is claimed that this holds if the prediction $\hat{x}_0(x_t)$ is compactly supported but I am not sure how this is possible even if the data distribution is compactly supported. Indeed, the marginals of the forward process $q_t$ are the result of the convolution of the data distribution with a Gaussian kernel so that it cannot be compactly supported. Since  $\hat{x}_0(x_t)$ contains the score, it cannot be compactly supported too. Next, assuming that the score is bounded is also not reasonable. Finally, assuming that $p(y; x^0)$ is bounded away from zero works but this does not hold for Gaussian linear inverse problems, which is quite unfortunate.

In short, both assumptions (b) and (c) are strong assumptions and it should be stated explicitely.

There is a typo in assumption (c); its not the gradient that should be continuous with bounded gradients.

**Limitations:**

See weaknesses.

---

> ### Author Rebuttal · Authors · 2023-08-10
>
> We thank the reviewer for their thoughtful comments and are glad they found the paper to be original, well-written and to have sound theoretical and numerical support.  We believe that our new experiments and clarifications below thoroughly addresses the weaknesses noted.
>
> **Computational cost.**  Compared to other approaches, we believe the proposed algorithm can present a favorable trade-off between computational cost and accuracy.  But we agree with the reviewer that the proposed algorithm is not the best idea when both the following hold: (1) repeated, fast generation at inference time is of primary concern and (2) designing and training an effective conditional model is an option.  We summarize some comparisons both from our new experiments and submission:
> * For the class-conditional ImageNet experiment described in our high-level response, we estimate the compute spent on training a noisy classifier required by Classifier Guidance was approximately 330 gpu hours ([they](https://github.com/openai/guided-diffusion/tree/main) report 300,000 training iterations with a batch size of 256, and in our hands one training iteration with a batch size of 8 takes 0.1244 GPU seconds).   By contrast, generation of a sample using TDS with 16 particles requires 6 GPU minutes, 0.03% of the conditional training time providing comparable accuracy.  Moreover, TDS does not require the engineering time to assemble a labeled dataset and design the time-dependent model.  However, because inference time with classifier guidance is lower, the total compute cost would lower if many (e.g. > 3,300) samples were to be generated.  The reviewer is correct that parallelizing across many particles is not as straightforward for ImageNet due to high dimensionality, but could in principle be scaled across multiple GPUs or computed in sequence (we used the latter approach for our experiments).
> * Relative to SMCDiff, in our experiments the efficiency improvement more than compensates for the cost of each particle.  The compute cost of TDS is roughly 2.2X higher  than that of SMCDiff, but in Figure H we show that TDS with 2 particles outperforms SMCDiff with even 64 particles. In this situation, TDS is >10x faster and provides better results.  The observation about effective sample size of SMCDiff relative to TDS is interesting.  We are surprised by this result given the worse empirical performance of SMCDiff by classification accuracy and do not have a good explanation for why these metrics do not agree.
>
> **Impact of dimension:** Our promising empirical results on MNIST (784 dimensional), protein design (\~600 dimensional, varying by test-case), CIFAR 10 (\~3K dimensional) and ImageNet (196K dimensional) suggest TDS can work well even on high dimensional problems.
> Notably, in the ImageNet case, despite that the particles lack diversity in global-level features, there remain variations in local patches (see Figure 2).
> Indeed these high-dimensional, empirical successes are not typical for SMC methods, which can have notoriously bad dimension dependence.  We suspect this good performance in high dimensions owes to the quality of the proposals distributions obtained with our twisting functions.
>
>
> **Suitability of the assumptions of Theorem 1:**
> We thank the reviewer for their thoughtful engagement with the assumptions of our theorem.  We will replace the word “mild” with a discussion of conditions, as we agree that our claim of “mildness” in the main text was overly blase. However, we disagree with the claim that our conditions “don’t hold in the simplest cases” and hope the points below will satisfy the reviewer and (once added to our revision) future readers.
> * **Compact support of $\hat x_0(x_t)$:** We expect the range of $\hat x_0(x_t)$ to exist within some compact set for two reasons.  First, it is common practice to truncate denoising predictions to within the range of the data; for example in image diffusion models by clipping values between the maximum and minimum pixel intensities values.  Second, because $\hat x_0(x_t)$ is trained to approximate $E[x_0 | x_t]$ which exists within the convex hull of the population distribution; for example, the (mean centered) protein structures have bounded support because they contain a finite number of atoms whose distances are constrained by fixed bond lengths, and so we always observe denoising predictions within the bounds of some maximum size. We do not assume that the $x_t$’s for $t>0$ have compact support (which indeed is not satisfied in general).
> * **Bounded score:** Our assumption is not that the (unconditional) score is bounded (which indeed would not be satisfied), but that the gradient of the log likelihood approximation $\nabla_{x_t} \log \tilde p(y|x_t)$ is bounded.  As we noted on line 601, this is our “strongest” assumption but will be satisfied if $\hat x_0(x_t)$ and $p(y;x_0)$ are smooth in $x_t$ and $x_0$ respectively.  Though this condition is difficult to check, we suspect this assumption will be satisfied for typical denoising networks and classification models when trained with regularization to encourage smoothness.
> * **Applicability to linear inverse problems:** The reviewer is correct that assumption c does not hold in these cases, and we will make this more explicitly.  We had separated the likelihood case into a separate first section (3.2) in which we presented our theorem, and described the inverse problems case subsequently for this reason.  Generalizing our theory to cover this case as well is of interest for future work, but requires moving away from the standard SMC theory – although SMC methods are commonly used with weight functions that are not bounded (as in the inverse problems case) most existing theory requires this assumption.
>
> We thank the reviewer for identifying our typo in our statement of assumption (c).   We will correct it.  Thank you very much again for your careful reading!

---

> > ### Comment · Reviewer_BFYS · 2023-08-15
> >
> > I would like to thank the reviewers for their thoughtful and honest answer! It is quite surprising, and quite unintuitive for me, to see that $K = 2$ particles are enough already. My concerns are addressed and I have decided to raise my score.

---

> > > ### Author Response · Authors · 2023-08-16
> > >
> > > Thank you for your reply.  We are glad to have addressed your concerns.

---

### Official Review · Reviewer_bGjC · 2023-07-07

**Soundness:** 3 good
**Presentation:** 3 good
**Contribution:** 3 good
**Rating:** 5
**Confidence:** 4

**Summary:**

The paper proposes an SMC algorithm to draw conditional samples form a diffusion model. Specifically, they wish to draw samples p(x0|y) given a diffusion model p(x0) and likelihood p(y|x0). Existing techniques to do so rely on expensive training of conditional diffusion models or heuristics which do not sample from the correct conditional distribution.

**Strengths:**

- Theoretically well-founded method. The paper does a good job of describing the flaws of commonly used methods for conditional generation and proposes a technique to target the true conditional distribution without needing task-specific training.
- State-of-the-art results for protein motif scaffolding, which is a very reasonable application of the proposed exact targeting of conditional distributions

**Weaknesses:**

- For some of the experiments (such as inpainting), a reasonable baseline to compare against would be that proposed in Section 3.1 of VDM (https://arxiv.org/abs/2204.03458). This performs conditional generation with an unconditional diffusion model using a heuristic that they show improves on naive guidance (although doesn't have theoretical guarantees like the proposed SMC method). A comparison against this would make it clearer when Twisted SMC is indeed the best option.
- Section 4.3 is hard to read and lacking in plots or concrete results. Adding something like Figure J from the appendix would make the results more convincing for readers who do not venture to the appendix to search for plots.
- How accurate is the inferred conditional distribution when less sampling steps are used? With progressive distillation or more modern ODE/SDE solvers it is now common to sample from diffusion models with tens of integration steps. The performance of SMC presumably degrades when less steps are used, since each individual step must be bigger. Can the authors comment on this or quantify this effect? E.g. how would Twisted SMC perform relative to the baselines if the number of diffusion steps used for motif-scaffolding was halved?

**Questions:**

See weaknesses.

**Limitations:**

Yes

---

> ### Author Rebuttal · Authors · 2023-08-10
>
> We thank the reviewer for their comments and are glad they found the paper to be theoretically well-founded, and that they appreciate the state-of-the-art results.  We believe we thoroughly address the noted limitations in the below.
>
> **VDM as a baseline:** As we noted in our high-level reply, the method we labeled as “guidance” exactly coincides with “reconstruction guidance” as proposed in VDM.  We thank the reviewer for revealing that this term was unclear, and will be certain it is clarified in our revision.
>
> **Section 4.3 results and readability:**  We appreciate the suggestion to provide more concrete results and will move Figure J into the main text to accomplish this as suggested.  Additionally, we will add a brief preamble paragraph to section 4.3 to summarize the evaluation and state-of-the-art results.
>
> **Accuracy with fewer steps:** This is a great question.  We comment first on empirics and then on methodological considerations. With 50 or 100 steps rather than 200 steps performance seems to degrade to varying extents depending on the particular motif (see Table below).  The performance drop is statistically significant (by Fisher exact test, p<0.05) in only one of three cases (3IXT).
>
> However, the interpretation of this result is complicated by the fact that changing the number steps (or analogously using a different ODE/SDE integrator) slightly modifies the sampling distribution of the unconditional model, and therefore also slightly modifies conditional distributions.  Consequently, exactly quantifying the impact on accuracy is ill-posed because it is relative to a moving target.  We expect however that, though one can apply TDS with any number of sampling steps, effective sample sizes may be worse when step-sizes are large.  In fact, we conjecture that the KL between intermediate targets is linear in the step size, and therefore the number of particles needed could be exponential.  However, we have yet to prove this result and leave it to future work.  We additionally note that the performance of other conditional generation methods may also in some cases degrade when fewer steps are used.
>
> *Table:* Number of motif-scaffolding successes (out of 50 runs) with different numbers of diffusion steps with 8 particles.
>
> | Motif       | 200 steps | 100 steps | 50 steps |
> | ----------- | --------- | --------- | -------- |
> | 1QJG        | 3 / 50    | 1 / 50    | 1 / 50   |
> | 3IXT        | 48 / 50   | 43 / 50   | 42 / 40  |
> | 5TRV_short  | 20 / 50   | 9 / 50    | 8 / 50   |
>
> These results above are from a preliminary exploration (conducted before submission) of the impact of the number of steps.  In response to the reviewers comments we intend to rerun these experiments to more comprehensively understand the impact of the number of steps.

---

### Official Review · Reviewer_UG6i · 2023-07-07

**Soundness:** 2 fair
**Presentation:** 2 fair
**Contribution:** 2 fair
**Rating:** 5
**Confidence:** 4

**Summary:**

This paper proposes a practical approach for achieving asymptotically exact inference from diffusion models through exact conditional sampling in terms of Sequential Monte Carlo (SMC) . This discovers the connection between SMC and diffusion models, and one of the key feature is to approximate the optimal twisting functions by som tractable twisting alternatives. The proposed method is extended to tackle inpainting, inpainting with degrees of freedom problems, and make diffusion model compatible on Riemannian manifolds. The experiments involve synthetic diffusion models, class-conditional sample generation on MNIST dataset, and motif-scaffolding problem, which is relevant to protein design.

**Strengths:**

1. The inspiration of the paper is good, which tries to achieve exact conditional sampling from diffusion models by making connection to SMC, and extend to Riemannian diffusion models.
3. (More like an) Ablation study in the experiment portion to examine whether the proposed method works better.

**Weaknesses:**

Based on understandings,
1. The novelty of this paper is not clear to me. The strong and close connection between diffusion models and the SMC technique is not crystal clear to me. Even though the reverse process with T diffusion steps from the diffusion models and sampling across T steps in terms of SMC could be related somehow, the significance of this connection is not apparent. The authors also point out that Classifier guidance[1] requires additional classifier training, but no computational cost and sample quality comparison is provided; Classifier-free guidance [2] requires classifier information as an additional input but again, none of the aforementioned result is presented and class information is still needed by the proposed TDS method (from my understanding but could be wrong). Equation 10, which is referred to as key insight in the paper, not been explained well, and actually has been proposed in [5] with even further clear explanation. The previous paper is not cited either. The title is also not informative enough and the emphasis on SMC is not evident.
2. Lack of real benchmark for the first two experiments, particularly the examination on the MNIST dataset. I am also concerned that the "Guidance" method performs significantly worse than the other four methods without any justification, and no pre-trained diffusion model is served as a benchmark here. Three experiments are all competing to themselves. It would be more convincing for the image synthesis experiments if they were conducted on more diverse and challenging datasets such as CIFAR-10, instead of solely relying on the MNIST dataset, which consists of grayscale images with only 10 classes.
3. Not sure why but the ``Related work`` is added in the appendix. MNIST class-conditional generation experiment part is in the paper, the inpainting is put in the appendix as well (which is mentioned in the ``abstract`` and ``conclusion``). I do not think it is pretty common but I would suggest to move them into the main paper from appendix.
4. There is a big room to improve for writing. Too many typos in both grammar and math expressions, not being cited properly, i.e., Tweedie's formula in the ``background``; unclear descriptions, i.e., the diffusion models described in the ``background`` and ``methodology`` are not referred to VE or VP diffusion models until the appendix, and mathematical notations appear without proper explanation, i.e. $\nu_t(x^{t:T})$ the first occurrence on line 94, where its meaning is not clarified. The lack of coherence between sentences and paragraphs further hampers the clarity of the paper.

[1]: Prafulla Dhariwal and Alexander Nichol. Diffusion models beat GANs on image synthesis. Advances in Neural Information Processing Systems, 2021.
[2]: Jonathan Ho and Tim Salimans. Classifier-free diffusion guidance. arXiv preprint arXiv:2207.12598, 2022.
[3]: Hyungjin Chung, Jeongsol Kim, Michael T Mccann, Marc L Klasky, and Jong Chul Ye. Diffusion posterior sampling for general noisy inverse problems. In International Conference on Learning Representations, 2023.

**Questions:**

1. ``Choosing weighting function $w_T(x^T) = w_t(x^t, x^{t+1}) = 1$ for $t = 1, \dots, T$'' constantly appearing in the paper to leave the proposal functions only in the equation. Are you assuming no resampling here and no particle is being discarded? If so, please make it more clear.
2. For weighting function on line 95, $w_k^t$ is a function related to $x_k^t, x_k^{t+1}$ and no $y$ is involved, so what is the point of not including $y$ here? When twisted weighting function is defined on equation 15, it is related to $y$ by $\tilde{p}_\theta(y | x^t)$ though.
3. On line 141, ``recall that $\hat{x_0}(x^t) = \mathbb{E}_{p_\theta}[x^0 | x^t]$ if $p_\theta$ is optimized to exactly match the true distribution $q$''. What if the assumption does not held perfectly? Is there any statistical analysis to justify this?
4. On equation 10, ``the approximated optimal twisting functions is $\tilde{p}_\theta(y | x^t) := p(y; \hat{x}_0(x^t, t))$''. Is there any analysis (if this idea is from [5], please ignore this question about analysis) or experiment result for not denoising $x^t$ back to $x^0$ for comparison?
5. Is the first experiment two-dimensional? What if it is high-dimensional?
6. What's the computational cost for various number of particles? Is there any variance reported for different number of particles for each experiment?

**Limitations:**

I would expect the authors address the weakness and questions aforementioned.

---

> ### Author Rebuttal · Authors · 2023-08-10
>
> We thank the reviewer for their detailed review and are glad they appreciate the inspiration for the method.  We hope the new clarifications and benchmarks described in the below provide improved support for the progress we have made in this direction.
>
> __Weaknesses:__
> 1. The first comment on weakness has several components that we address independently:
>  - **Significance between the connection between SMC and diffusion models.** The significance is that it allows us to use heuristic approximations to conditional sampling in diffusion models to implement a practical and asymptotically accurate conditional sampling procedure (See e.g. lines 51-57).
>  - **Comparison to conditional approaches.** We note that class information is required for TDS since the goal is to generate samples given a class. The advantage of TDS is that it does not require additional training of a classifier on noisy inputs diffused at various steps (as in classifier-guidance), or a conditional diffusion model trained on class information (as in classifier-free guidance). TDS can operate on a pre-trained unconditional diffusion model and a standard classifier trained on clean inputs. See also our response to weakness (2).
>   - **Novelty of equation 10.** With regard to Eq.10, the insight is that such approximations (previously suggested in [5] and also [15,21]) can enter an SMC procedure as a “twisting function”. While we have discussed this connection in line 151 as well as in related works line 519, we will make it more clear in the main text.
>  - **Title.** We appreciate the suggestion to change the title, as we recognize that the substance of our contributions build more heavily on existing SMC methods than one might have thought.  **Our new proposed title is:  "Practical and Asymptotically Exact Conditional Sampling in Diffusion Models"**
> 2. At the reviewer’s suggestion we now include applications to CIFAR10 and ImageNet, and compare to a conditionally-trained classifier guidance baseline, and believe the results (described in the high-level response) strengthen the paper.
> 3. We appreciate the suggestion and will move the discussion of related work into the main text.  We will either additionally move the image inpainting results into the main text (space allowing) or strike the references to these experiments from the abstract and conclusion.
> 4. We appreciate the comments on writing and will add appropriate references and discussions of the connections to VE and VP models to the main text.
>
> __Questions:__
> 1. Yes. When weights are uniformly one, one can skip resampling steps to reduce the variance of the procedure (see footnote on page4).
> 2. We thank the reviewer for pointing out this omission. We have left “y” as an implicit input, but will make its omission explicit in the revision.
> 3. Yes, this is a great question.  If it does not hold perfectly, the distribution of the samples returned by TDS has some deviation from the exact conditional distribution.  A primary contribution is to show that this deviation may be reduced arbitrarily by increasing the number of particles (Theorem 1).
> 4. This approximation does indeed come from reference [5] (inheriting from [15] before it).  See also our response to weakness 1.  Not denoising back to x_0 is an interesting idea, but we suspect it would not provide a sensible result because noisy states would be very dramatically out of distribution for most likelihoods as compared to the denoised predictions.
> 5. The first experiment is indeed two dimensional.  This experiment was chosen to be as small as possible to be illustrative; we intended the 784-dimensional MINST experiments to illustrate a higher dimensional toy problem, and hope that our ImageNet experiments further address concerns about higher dimensional problems.
> 6. The compute cost increases linearly with the number of particles, and the inverse variance increases linearly with the number of particles.  This rate is a consequence of standard SMC results and can be seen from the slope of -1 (on log-log-scale) in our simulation experiments.

---

> > ### Comment · Reviewer_UG6i · 2023-08-14
> > **Response to Authors**
> >
> > I would like to thank the authors for their answers and the additional information. Most of my questions and concerns were touched and addressed. For the additional experiments on ImageNet, I can see a big improvement on classification accuracy when the classifier guidance can already achieve 99% accuracy; however, it is noteworthy that the quality and diversity of the generated samples, as indicated by the FID and Inception scores, have notably decreased.. These induced to two questions: 1) Is it justifiable to sacrifice sample quality and diversity to improve classification accuracy by applying TDS? This is without considering the increased time consumption resulting from the higher number of particles. 2) I may still not fully understand it, but can it be considered a valid assumption to employ TDS performed upon a pre-trained classifier guidance, particularly when the training a classifier is time-intensive?
> >
> > I would be happy to update my score once my concern is addressed.

---

> > > ### Author Response · Authors · 2023-08-16
> > >
> > > Thank you for your reply.  We reply to the questions and then expand on our FID results, which we feel may appear artificially bad:
> > > (1) For class-conditional generation, if one has access to an already-trained noise-level-dependent classifier, we suspect it is not worth running TDS.  TDS is not meant to address this setting.
> > > (2) Yes. In many settings one may have access to a classifier trained (e.g. by someone at a different organization) on _only noise-free data_, and wish to guide their generations.  In such settings, running TDS would be possible without requiring either (a) further training or even (b) access to a labeled dataset.  By contrast, both (a) and (b) are requirements of classifier guidance.
> > >
> > > We next clarify our FID results.  You are correct that we reported lower FID scores with TDS relative to classifier guidance.  However we suspect this decrease owes primarily to a decrease in diversity, rather than a decrease in quality.  In particular, our evaluation was roughly matched on inference-time computation cost; for classifier guidance and TDS (P=1) we generated 16 _independent samples_ per class, whereas for TDS (P=16) we generated 16 _dependent_ samples for each class due to resampling (in total, 16,000 images were generated for each method).  Using TDS P=16 increases quality relative to P=1 (as suggested by the visual results, classification accuracy, and inception score) but the dependence between particles lessens diversity and thereby increases FID. To demonstrate this, we have run TDS (P=16) a second time and merged the samples. In this case, we obtain an FID score of 22.68 and suspect we would see further improvements with additional independent runs of TDS (but have not done so due to the computational resources available on short notice). To match the previous evaluation size, we also obtain the average FID score of 23.75 (with std of 0.17) evaluated on 16k randomly selected samples out of 32k combined samples over 10 random selections. We note that these FID scores are lower than that of samples from the unconditional model (26.2). In practice, one can increase the diversity of TDS samples by (1) running TDS for multiple individual runs, or (2) reducing the resampling frequency so particles can be less dependent on each other (in our experiment we resample at every step).

---

> > > > ### Comment · Reviewer_UG6i · 2023-08-21
> > > > **Response to Authors**
> > > >
> > > > Thank you for authors’ response again. I have raised my score.

---

### Official Review · Reviewer_8aww · 2023-07-19

**Soundness:** 4 excellent
**Presentation:** 3 good
**Contribution:** 3 good
**Rating:** 5
**Confidence:** 4

**Summary:**

This paper addresses the challenge conditioning in unconditionally-trained diffusion models. The most successful approaches often require explicitly training on conditional data. This paper frames sampling from such conditionals as an SMC procedure and proposes Twisted Diffusion Sampler (TDS), a method derived from the twisting technique in SMC literature. TDS uses a classifier trained on the clean, non-noisy data to construct a sub-optimal proposal distribution for sampling from the diffusion process. This proposal is then used in the twisted SMC framework to get weighted particles that more closely approximate the conditional distribution of interest.

The authors prove that their proposed SMC method targets the correct distribution and is asymptotically exact. Furthermore, they show how to extend TDS to work for various inpainting problems and Reimannian diffusion models.

Finally, the authors empirically verify TDS's correctness on a simple synthetic diffusion model with tractable score functions. Then they show its effectiveness on more realistic tasks of class-conditional MNIST image generation and inpainting. They lastly show TDS achieves state-of-the-art results on motif-scaffolding problem in protein design.

**Strengths:**

The central contribution of the paper is introducing the twisted SMC framework for sampling from diffusion models. While the twisted SMC is not novel, applying it to diffusion models and demonstrating use cases is significant. Moreover, the extension of TDS to some settings other than class-conditional framework was very interesting to me.

In terms of clarity, this paper is mostly clearly written. There are some typos and clarifications required that I have listed in the questions section.

**Weaknesses:**

I think the main weakness of the paper lies in the experimental section. First, the experiments are rather small scale. It would be nice to see how their method perform on larger scales standard tasks such as class-conditional CIFAR or ImageNet generation. Second, I expected to see more baselines in the experiments. The paper is motivated by pointing out that many existing methods require additional training. I believe it such methods should be included as baseline. For example, classifier-guidance with a proper classifier trained on noisy data or classifier-free guidance would be good baselines to compare against. Admittedly, they might outperform TDS, particularly with smaller K. However, it would be a good upper bound to have. Moreover, it gives insights on the tradeoff between a large K or training a separate model.

**Questions:**

1. I am not sure if what the authors refer to "reconstruction guidance" is correct. Quoting from the related work section "the use of denoising estimate $\hat{x}_0(x^t, t)$ to form an approximation $\tilde{p}_\theta(x^t | x^{t+1}, y)$ to $p_\theta(x^t | x^{t+1}, y)$ is called _reconstruction guidance_." I am not sure if this is correct. As far as I know, reconstruction guidance is proposed in [1] and improves the replacement method [2] which is for inpainting applications. Moreover, the other two citations are [3,4] (citation numbers 5, 25 in the paper) which are unrelated to reconstruction guidance (this might be related to the question 3 below though)
2. In the related work section (App. B), on line 523, there is a "[33]" citation which does not exist. The reference 29 in the paper ([5] here) seems to be what it meant to refer to. This paper, however, does not do reconstruction guidance as far as I can tell.
3. Looking at the rest of the related work section, seems like most citation numbers are incorrect. For example citations 26 and 18 on lines 524 and 528, respectively do not seem to refer to the correct paper. I suggest double-checking the citations.
4. In section 3.3, it is not clear to me how Eq. (18) is derived and whether it is a convenient likelihood that often works in practice or it is a general result. I did not see an explanation in the appendices either. I think it would be helpful to clarify this.
5. In the experiments section
   1. "Guidance" is defined as "TDS" with one sample. If I understand it correctly, it would be equivalent to "TDS-IS" with one sample. Is this correct? If so, I expect it to be outperformed by "TDS-IS" with more samples while the middle and right panel of Figure 1 shows otherwise. Additionally, in Figure 2(a), I do not understand why would "TDS-IS" show less diversity than "guidance". Why would more particles destroy the diversity?
   2. I am wondering if the authors have insights on why in Figure 2(b) ESS jumps back up very close to t=0.
6. Minor issues
   - Section 3 and subsection 3.2 have the same name.
   - The background section on diffusion models explains diffusion processes without scaling (e.g. the variance-preserving process).It is worth mentioning in the background section that not all diffusion models follow this exact framework.
   - I think it's nicer if Eq. 11 is written to define $\tilde{p}_\theta(x^{t-1} | x^{t}, y)$ instead of $\tilde{p}_\theta(x^t | x^{t+1}, y)$ to be similar to Eq. 3.
7. Typos
   - Line 9: the -> that
   - Line 99: provide -> provides
   - Line 148: eq. -> Eq.
   - Line 153: I might be wrong, but I guess $s_\theta(x^t, y) = \nabla_{x^t} \log q(x^t, y)$ should actually be $s_\theta(x^t) = \nabla_{x^t} \log q(x^t)$. Otherwise, the assumption of $\tilde{p}_\theta(y | x^t) = p_\theta(y | x^t)$ becomes unnecessary for Eqs. (13,14).
   - Line 157: $w_T(x^T) := \tilde{p}_\theta(x^T)$ -> $w_T(x^T) := \tilde{p}_\theta(y | x^T)$
   - Linr 218: sytematic -> systematic
   - Lien 570: systematica -> systematic
   - In the second line of equations under line 578: $p(y | x^{t'})$ -> $p(y | x^{t})$

[1] Jonathan Ho, Tim Salimans, Alexey A Gritsenko, William Chan, Mohammad Norouzi, David J Fleet.
"Video diffusion models".
NeurIPS 2022.

[2] Yang Song, Jascha Sohl-Dickstein, Diederik P Kingma, Abhishek Kumar, Stefano Ermon, Ben Poole.
"Score-based generative modeling through stochastic differential equations".
ICLR 2020.

[3] Hyungjin Chung, Jeongsol Kim, Michael T Mccann, Marc L Klasky, Jong Chul Ye.
"Diffusion posterior sampling for general noisy inverse problems".
ICML 2023.

[4] Jue Wang, Sidney Lisanza, David Juergens, Doug Tischer, Joseph L Watson, Karla M Castro, Robert Ragotte, Amijai Saragovi, Lukas F Milles, Minkyung Baek, et al.
"Scaffolding protein functional sites using deep learning".
Science 2022.

[5] Guanhua Zhang, Jiabao Ji, Yang Zhang, Mo Yu, Tommi Jaakkola, and Shiyu Chang. "Towards coherent image inpainting using denoising diffusion implicit models".
arXiv preprint, 2023.

**Limitations:**

The main limitation of the proposed method is the additional sampling cost due to multiple SMC particles which is mentioned in the discussion section of the paper. Otherwise, I do not see other limitations.

---

> ### Author Rebuttal · Authors · 2023-08-10
>
> We thank the reviewer for their very careful and detailed review.  We are glad they found it to be significant and interesting.  We found the questions and weaknesses noted helpful and believe we have addressed them below.
>
> **Improving the experimental validation:** As noted in the response to all reviewers, we have implemented the suggestion to include class-conditional CIFAR and ImageNet generation, and find TDS to be competitive with classifier guidance (albeit at larger inference-time cost).  As the reviewer suggested, classifier guidance does indeed outperform TDS with just a single particle but performs similarly by Classification accuracy (see table 1 in high-level reply).
>
> **Replies to questions:**
> 1. Reconstruction guidance uses the reconstruction estimate $\hat x_0(x_t)$ as a proxy for the true conditional mean $E[x_0|x_t]$. Eq 7 in VDM paper [15] demonstrates the use of reconstruction guidance gradient obtained by backpropagating through the denoising network. In the inpainting case, TDS with P=1 coincides with the reconstruction guidance. We will make this connection more clear in the revised paper.
> 2. The reviewer is correct that these references were wrong. We apologize for our mistake, which resulted from an accidental reordering of references between when we compiled and submitted our main text and our appendix files.
> 3. See (2) above.
> 4.  We thank the reviewer for pointing out that Eq.18 was unclear.  The final proposal and weights in Eq.18 are chosen to ensure that the final target is the exact conditional distribution of interest.  The intuition is that as defined in Eq.18,  tilde $p (x_0 | x_1, y) = p (x_0 | x_1, y)$, and as a result reweighting is not needed.  By substituting equation 18 into the expression for $\nu_0$ in equation 16, we obtain the desired target.  We will clarify this in our revision.
> 5. This question touches on both TDS-IS and ESS plots:
>   - *TDS- IS.*  We thank the reviewer for pointing out this surprising behavior.  We have investigated it and discovered that it owes to a bug in the implementation of our simulations; in short, we used an incorrect and unstable implementation of the final step weights. Our attached PDF presents the results for inpainting (Figure 4) where we find that the results agree with the reviewer's expectations.  Our revision will include corrected version of the remaining three simulations with this bug-fix.
> As for why TDS-IS provides lower diversity as compared to guidance in Figure 2a, this is in essence a bias-variance trade-off.  TDS-IS explicitly weights and resamples based on the likelihood approximation at the last step. This resampling replicates some samples and eliminates others, which reduces diversity giving some samples that are identical to one another. By contrast, guidance generates independent samples and does not use importance weighting and resampling, hence resulting in a higher diversity but worse average quality.
>  - *ESS jumps near t=0.*  This is a good question which we have wondered about as well.  We report additional ESS traces in Figure 5 showing that this jump happens irregularly; we see it regularly for MINST, sometimes for CIFAR10, but usually not on ImageNet (Figure 5A, 5B, and 5C).  Mechanically, the drop in ESS implies a large discrepancy between the final and intermediate target distributions.  We suspect such discrepancies might arise from irregularities in the denoising network near t=0, and will add a brief discussion of this behavior and its implications on particle diversity and the value of truncating resampling in the final steps.
> 6. This question touches on section titles, diffusion model background and time-indices.
>  - *section 3 and 3.2 titles.*  We thank the reviewer for pointing this out.  We will provide more specific headings: “3: Twisted Diffusion Sampler: SMC sampling for diffusion model conditionals" and “3.2: Twisting functions and convergence for smooth likelihoods”.
>  - *diffusion model background.*  We thank the reviewer for pointing out this confusion.  We omit the scaling factor to simplify demonstration, and include the discussion of  variance preserving and variance exploding models in appendix D. We will add a reference to the appendix in the revised background section.
>  - *time indices.*  We appreciate the suggestion.  However we prefer to keep this choice of indexing to better agree with the indexing we use for sequential Monte Carlo.  Unfortunately, this cannot instead be resolved by changing the indexing of diffusion models in equation (3) without requiring $x_{t+1}$ as an argument of the denoiser and score model throughout.
> 7. We thank the reviewer for identifying these typos.  These were all correctly identified and will be resolved in the revision.

---

> > ### Comment · Reviewer_8aww · 2023-08-21
> > **Response to Authors**
> >
> > I appreciate the authors' effort to conduct further experiments and their clarifications. Most of my questions are answered and I will raise my score. However, the new results do not demonstrate a clear advantage from the techniques introduced. In particular, the scores of TDS with one particle (which is equivalent to "naive" guidance) very close to TDS with 16 particles. The exception is inception score where TDS (P=16) shows a higher score suggesting more sample diversity, but the diversity of reported images is very limited.
> >
> > Nonetheless, I still believe the proposed method and the details of implementing it in different application is elegant and fruitful to the community.

---

### Author Rebuttal · Authors · 2023-08-10

We thank the reviewers for their detailed comments and suggestions. We are pleased that the reviewers found our method to be “practical”, “original”, “theoretically well-founded”, and to give “state-of-the-art” results in protein design. We believe we have addressed all suggested weaknesses, and have improved the submission both by adding several new experiments, and through revisions to the text.  Notably we apply TDS to 256x256 ImageNet class-conditional generation and find TDS a viable alternative to Classifier Guidance which requires expensive training of a classifier on noisy inputs.

Before describing these changes we provide additional high-level context and summarize our contributions.

Our submission proposes an algorithm, TDS,  for accurately estimating conditional distributions implied by diffusion models useful in cases when it is desirable to trade off sampling-time with accuracy. For example, the development of protein-based drugs might involve (1) developing and training a diffusion model of protein structures, (2) sampling candidate protein structures from the model, and (3) experimentally and clinically validating candidate proteins.  While steps (1) and (3) might require years, sampling (2) takes only seconds.  The proposed algorithm would apply in this second step. On several full-scale protein tasks, we show that TDS significantly outperforms the previous state of the art, a conditionally-trained diffusion model [26].  Moreover, we show the algorithm allows one to target conditional distributions to which previous methods do not even apply. In addition, we evaluated TDS on synthetic problems and well-studied computer vision benchmarks.

We now move to shared concerns with discussions of new experiments with figures in the attached PDF.

__Higher-dimensional image problems and comparison to baselines.__
Reviewers 8aww, UG6i and BFYS noted that our submission had not explored performance on high-dimensional problems.  To this end we have included additional experiments on class-conditional generation on CIFAR10 and 256x256 ImageNet.
- ImageNet: we compare TDS (with # of particles P=1,16) to Classifier Guidance (CG) using the same unconditional model from [here](https://github.com/openai/guided-diffusion/tree/main). CG uses a classifier trained on noisy inputs. For TDS, we use the same classifier evaluated at timestep = 0 to mimic a standard trained classifier. We generate 16 images for each of the 1000 class labels, using a guidance scale of 10 and 100 sampling steps. Notably, given a fixed class, TDS(P=16) generates correlated samples in a single SMC run, and TDS(P=1) and CG generate 16 independent samples.
TDS can faithfully capture the class and have comparable image quality to CG’s (Figure 2, class: bramblings), although with less diversity than CG and TDS(P=1). Figure 3 shows more samples given randomly selected classes. We also reported results of the unconditional model from [1] that are evaluated on 50k samples with 250 sampling steps (Table 1). TDS and CG provide similar classification accuracy. TDS has similar FIDs compared to the unconditional model and better inception score. CG’s FID and inception score are better than TDS. We suspect this difference is attributed to the sample correlation (and hence less diversity) within particles in a single run of TDS(P=16).
- CIFAR10: we ran TDS (P=16) with guidance scale = 1 and 100 sampling steps (Figure 1) using diffusion model from [here](https://github.com/openai/improved-diffusion) and classifier from [here](https://github.com/VSehwag/minimal-diffusion/tree/main). TDS generates faithful and diverse images.  However, we found TDS can occasionally generate off-the-manifold samples for the class ‘truck’ (Figure 1F).

__Reconstruction guidance, video diffusion models (VDM) and baselines terminology.__ Reviewer bGjC suggested comparing TDS to the reconstruction guidance approach presented in “Video Diffusion Models” [15], and reviewer 8aww identified that our description of this baseline (first introduced in [15]) was unclear.  We compared TDS to reconstruction guidance in our submission under the name “guidance”, a choice of terminology we will clarify in our revision.

In brief, we used “guidance” to describe a generalization of “reconstruction guidance” to classification problems and inpainting with degrees of freedom (App. B); in the inpainting case, it exactly coincides with reconstruction guidance [15].  Our approach may be seen as wrapping this heuristic approximation in an SMC sampler to improve accuracy by adding additional compute (lines 53-55).  See our reply to reviewer bGjC for details.

__Compute Cost.__ We will include the below discussion in our revised manuscript.
- Classifier guidance vs TDS:  In Classifier Guidance [8] one trains a noise-level dependent classifier.  One therefore incurs a large, up-front computational cost to train the classifier. It amortizes the inference problem and allows conditional samples to be generated in a single trajectory and so is fast at inference time.  By comparison, TDS demands compute that is (1) linear in the number of particles used and (2) higher by a constant factor due to backpropagating through the denoising network.  In the ImageNet model, TDS takes 0.34s to generate one particle at a time step, and Classifier Guidance takes 0.15s (results averaged over 100 samples on a V100 GPU). So TDS requires 120% more gpu time in this instance. Notably, training a noisy classifier required by Classifier Guidance can take around 330 gpu hours.

- TDS vs RFDiffusion on motif-scaffolding:  TDS(P=8) is faster than the state of the art baseline (RFDiffusion).  On a 100 residue test case (1QJG) on an A4000 gpu, run-time was 80 seconds for TDS, as compared to 150 seconds for RFdiffusion.  In both cases the models used 200 diffusion time-steps.  The slower speed of RFdiffusion owes to its use of a larger neural network.

---

### Decision · Program_Chairs · 2023-09-21

**Decision:**

Accept (poster)

**Comment:**

I am thrilled to convey my recommendation for the acceptance of your paper to be presented at NeurIPS. Your work has impressed both the reviewers and myself with its quality and significance, and I am confident that it will make a valuable contribution to the NeurIPS conference.

Your responsiveness to the feedback provided during the review process reflects your commitment to advancing the field and strengthening your contribution. Your willingness to engage with the reviewers' insights demonstrates your dedication to delivering a high-quality presentation that will resonate with the conference attendees.

NeurIPS is a prestigious platform that attracts the brightest minds in the field, and your paper's acceptance adds to the conference's reputation for excellence. I am genuinely excited to see your work presented and discussed among peers who share your passion for pushing the boundaries of knowledge.